# Tryptophan fuels MYC-dependent liver tumorigenesis through indole 3-pyruvate synthesis

Niranjan Venkateswaran[1,9], Roy Garcia[1,9], M. Carmen Lafita-Navarro[1,9], Yi-Heng Hao[1,9], Lizbeth Perez-Castro[1,9], Pedro A. S. Nogueira[1,9], Ashley Solmonson[2], Ilgen Mender[1], Jessica A. Kilgore[3], Shun Fang[1], Isabella N. Brown[1], Li Li[1], Emily Parks[1], Igor Lopes dos Santos[1], Mahima Bhaskar[1], Jiwoong Kim[4], Yuemeng Jia[2], Andrew Lemoff[3], Nick V. Grishin[5,6], Lisa Kinch[5,6], Lin Xu[4], Noelle S. Williams[3], Jerry W. Shay[1,7], Ralph J. DeBerardinis[2,6,7], Hao Zhu[2,7,8] & Maralice Conacci-Sorrell[1,7,8] ✉

Cancer cells exhibit distinct metabolic activities and nutritional dependencies compared to normal cells. Thus, characterization of nutrient demands by individual tumor types may identify specific vulnerabilities that can be manipulated to target the destruction of cancer cells. We find that MYC-driven liver tumors rely on augmented tryptophan (Trp) uptake, yet Trp utilization to generate metabolites in the kynurenine (Kyn) pathway is reduced. Depriving MYC-driven tumors of Trp through a No-Trp diet not only prevents tumor growth but also restores the transcriptional profile of normal liver cells. Despite Trp starvation, protein synthesis remains unhindered in liver cancer cells. We define a crucial role for the Trp-derived metabolite indole 3-pyruvate (I3P) in liver tumor growth. I3P supplementation effectively restores the growth of liver cancer cells starved of Trp. These findings suggest that I3P is a potential therapeutic target in MYC-driven cancers. Developing methods to target this metabolite represents a potential avenue for liver cancer treatment.

Through the activity of oncogenes, cancer cells develop an increased ability to obtain nutrients and efficiently utilize them for biomass production and cell growth[1]. Recent studies explored the efficacy of limiting single dietary components in cancer treatment[2–4]. For example, methionine deprivation reduces the growth of colon cancer cells[5–7], and asparagine deprivation prevents the growth of cancer cells by altering mitochondrial function[3]. Furthermore, diets restricted in serine and glycine reduce cancer cell growth through the accumulation of deoxysphingolipids[2–4]. Because the different types of cancer vary in their metabolic activity, preferred energy source, and nutritional dependencies, characterization of nutrient demands by individual tumor types has the potential to identify their specific

[1]Department of Cell Biology, University of Texas Southwestern Medical Center, Dallas, TX 75390, USA. [2]Children's Medical Center Research Institute at University of Texas Southwestern Medical Center, Dallas, TX 75390, USA. [3]Department of Biochemistry, University of Texas Southwestern Medical Center, Dallas, TX 75390, USA. [4]Quantitative Biomedical Research Center, Peter O'Donnell Jr. School of Public Health, University of Texas Southwestern Medical Center, Dallas, TX, USA. [5]Department of Biophysics, University of Texas Southwestern Medical Center, Dallas, TX 75390, USA. [6]Howard Hughes Medical Institute, University of Texas Southwestern Medical Center, Dallas, TX 75390, USA. [7]Harold C. Simmons Comprehensive Cancer Center, University of Texas Southwestern Medical Center, Dallas, TX 75390, USA. [8]Hamon Center for Regenerative Science and Medicine, University of Texas Southwestern Medical Center, Dallas, TX 75390, USA. [9]These authors contributed equally: Niranjan Venkateswaran, Roy Garcia, M. Carmen Lafita-Navarro, Yi-Heng Hao, Lizbeth Perez-Castro, Pedro A. S. Nogueira. ✉e-mail: Maralice.ConacciSorrell@UTSouthwestern.edu

vulnerabilities that can be safely manipulated through dietary or pharmacologic interventions to target the destruction of cancer cells while minimally affecting healthy tissues.

Hepatocellular carcinoma (HCC) is the third leading cause of cancer mortality worldwide[8]. Despite therapeutic efforts, liver cancer is a highly refractory disease with a five-year survival rate of 30%. Hyperactivation of the WNT/β-catenin pathway is a major contributor to the pathogenesis of liver cancer[9]. One of the major targets of the WNT pathway is the universal oncogene *MYC*. MYC is aberrantly activated in patients with liver cancer and is associated with poor prognosis[10,11]. In mice, MYC is the most potent driver of HCC[12]. Interestingly, MYC promotes the uptake of Trp through increasing the expression of the Trp transporters SLC1A5 and SLC7A5 to drive cell growth in cultured cancer cells[13,14].

Trp is a multifunctional essential amino acid that can be incorporated into new proteins, be converted into serotonin, and be metabolized by the Kyn pathway[15,16]. The first step of the Kyn pathway is catabolized by 3 enzymes: indoleamine 2,3-dioxygenase 1 (IDO1), IDO2, and tryptophan 2,3-dioxygenase (TDO2)[17,18]. This produces the intermediate N-formyl kynurenine, which is converted into Kyn by arylformamidase (AFMID). Kyn can be further metabolized into kynurenic acid, cinnabarinic acid, xanthurenic acid, picolinic acid, quinolinic acid, and NAD[+19-21]. The amounts and activity of different enzymes define the production rate and stability of specific Trp metabolites[16]. We previously demonstrated that Kyn levels are increased in colon cancer cells and in 90% of colon cancer tumors via increased Trp uptake and elevated expression of Trp-metabolizing enzymes[13]. Kyn also causes T-cell inhibition and prevents cancer cell clearance[20,22-25]. While IDO levels are elevated in certain tumors, several inhibitors that are designed to block the activity of IDO1 have failed in clinical trials[26], suggesting that a more complex and potentially tissue-specific role for Trp and its metabolites may be involved in tumor growth.

Trp was also recently shown to generate indole 3-pyruvate (I3P) via the activity of interleukin 4-induced 1 (IL4I1)[27]. Like Kyn, I3P drives proliferation as a ligand for the transcription factor aryl hydrocarbon receptor (AHR)[13,27], which regulates genes involved in cell growth[28,29]. The importance of I3P pathway in cell autonomous and immune-modulatory pathways is just beginning to be explored. Here, we demonstrate that liver tumors initiated by the transgenic expression of MYC take up larger amounts of Trp than normal liver tissue and generate I3P, which functions as an oncometabolite that drives tumor growth.

## Results

### MYC-driven liver tumors exhibit an increase in Trp uptake

In the liver, Trp can be incorporated into proteins or other metabolites[16] (Fig. 1a). Using RNA-seq to compare normal livers or liver tumors driven by the overexpression of MYC as a transgene (Fig. 1b), we found that the expression of the Trp transporters SLC1A5 and SLC7A5, but not the Trp-metabolizing enzymes (TDO2, IDO2, AFMID, AADAT, KYNU, QPRT), were elevated in tumors (Fig. 1c). IDO1 expression was not measurable in this RNA-seq experiment. Examining a previously published dataset of MYC-induced liver tumors[30], we confirmed that SLC1A5 and SLC7A5 were induced by MYC and that turning MYC off rapidly abrogated their expression (Fig. 1d), thus indicating that SLC1A5 and SLC7A5 are likely direct MYC targets in liver cancer as shown for colon cancer[13].

In line with the elevated expression of SLC1A5 and SLC7A5 in liver cancer, Trp levels were elevated in livers of mice upon MYC upregulation, as measured by tandem mass spectrometry (LC-MS/MS) (Figs. 1e, S1a). In the same samples, the Trp metabolites Kyn and kynurenic acid (KA) were decreased but Cinnabarinic acid (CA) and serotonin was not altered (Figs. 1f–i, S1a). To quantify Trp uptake by liver cells in vivo, we infused control and mice carrying MYC-driven

liver tumors with Trp isotopically labeled in all carbons ($^{13}$C-Trp) by intravenous injection using experimental conditions previously established for glutamine[31,32] (Fig. 1j). The livers were dissected 1 h after injection and subjected to LC-MS/MS. This experiment revealed that $^{13}$C-Trp from the bloodstream made a larger contribution to the intracellular Trp pool in liver tumors than normal tissues, consistent with elevated uptake in tumors (Figs. 1k, S1b). Circulating $^{13}$C-Trp made a smaller contribution to the products of Trp, including Kyn, and KA, with serotonin levels below detection limit (Fig. 1 l, m). Other metabolites downstream were not measurable.

NAD$^+$ and NADP$^+$ were also reduced in MYC-driven liver tumors (Fig. 1n, o). NAD$^+$ steady state levels (Fig. 1n) and levels generated from $^{13}$C-Trp (Fig. 1o) were reduced in MYC-ON livers in agreement with the reduction in the expression of the enzymes necessary for their production (Fig. 1p, q). Enzymes involved in generating NAD$^+$ via pathways that are independent of Trp were also downregulated in mouse liver tumors (Fig. 1p, q, Fig. S1c). This finding supports the previous model of HCC driven by the prefoldin-like chaperone (URI), which displayed a reduction in Trp metabolites and Trp-metabolizing enzymes in parallel with the acquisition of growth-promoting pathways including phosphorylation of S6 kinase (also observed here, Fig. 1q)[33].

### Characterizing physiological effects of synthetic diets containing low or no Trp

To test the importance of increased Trp for liver cancer growth, we used synthetic diets that contained established amounts of all nutrients but with Trp reduced or absent (Table S2 for diet composition, Fig. 2a). First, we tested these diets in normal mice. It took 21 days for the No-Trp diet to cause a significant reduction in Trp levels in the liver. In contrast, the reduced Trp diet did not affect Trp levels in the liver or in circulation (Figs. S2a, b, 2b–g). The No-Trp diet caused a 40% reduction of Trp levels in the liver (Fig. 2b) and a ~10-fold reduction in the serum (Fig. 2e). As expected, the levels of Kyn and serotonin were also reduced in the livers and circulation of mice fed a diet lacking Trp (Fig. 2c, d, f, g). Trp-reduced diets had more modest effects. Trp was the only amino acid that was significantly low in the livers (Fig. 2h) and circulation (Fig. 2i) of mice fed the No-Trp diet. Interestingly and for reasons not yet determined, aspartate, glutamate, and serine in the livers (Fig. 2h) and methionine in the serum (Fig. 2i) were significantly higher in mice starved of Trp. Trp starvation resulted in a reduction in body weight in the No-Trp mice compared to mice fed the control diet (Fig. 2j); however, similar amounts of food were consumed by both groups (Fig. 2k–m).

Interestingly, animals fed No-Trp diet lost predominantly fat mass (Fig. 2n, o) while preserving lean mass (Fig. 2p, q). The weight loss caused by Trp starvation was fully rescued by feeding the No-Trp mice the control diet (Fig. 2r), suggesting that no permanent effects on body weight were caused by transiently altering dietary levels of Trp. Trp starvation had modest effects on respiration rate, (Fig. 2s). These results also suggest that side effects of short-term Trp starvation are limited and reversible.

### Trp deprivation reduces the growth of MYC-driven liver tumors

To test the importance of Trp for the initiation of MYC-driven liver cancer, we compared mice overexpressing MYC in the liver that were fed with control, Low-Trp, or No-Trp diets for 21 days (Fig. 3a). The mice fed the control diet died between days 47–55 of age, and the mice fed Trp-reduced diet survived longer (Fig. 3b). Mice fed the No-Trp diet had no or small tumors and were all alive on day 67 (Fig. 3b). Per IACUC recommendation, all animals were sacrificed on day 67 due to >20% weight loss. Liver size and weight were lower in the mice fed the No-Trp diet than in the mice fed the control diet. The liver weight of mice with MYC-driven tumors reached 40% of the body weight. However, the liver weight of mice fed the No-Trp diet was around 10% of their body

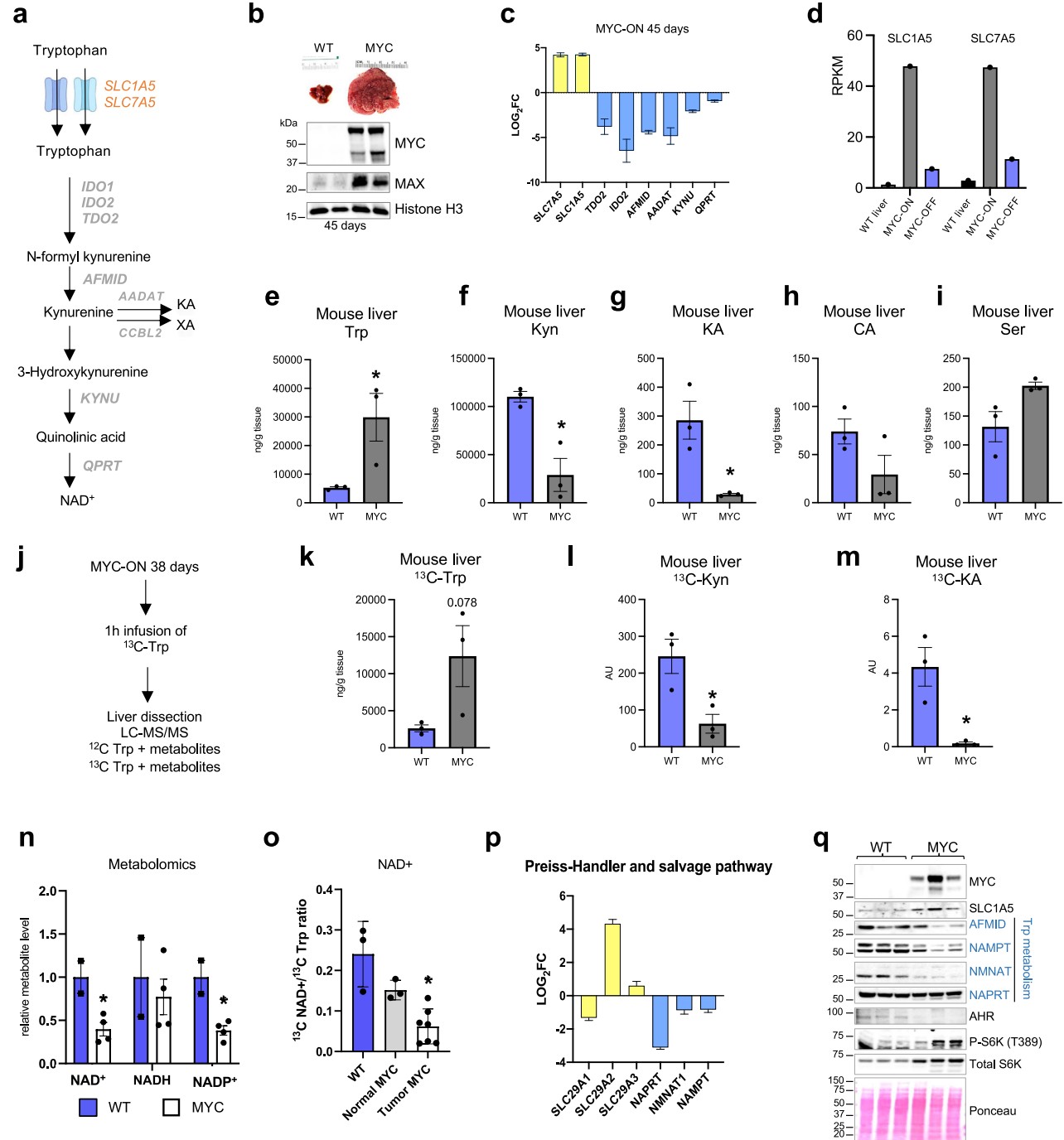

**Fig. 1 | MYC-driven liver tumors exhibit an increase in Trp uptake. a** The Kyn pathway and its enzymes. kynurenic acid (KA), xanthurenic acid (XA). **b** Relative size comparison of WT and MYC-expressing livers and Western blots of liver protein lysates from WT liver and liver in which MYC was turned on for 45 days. **c** Relative mRNA expression of Trp transporters and Trp-metabolizing enzymes plotted from RNA-seq of WT liver (*n* = 2) and liver in which MYC was turned on for 45 days (*n* = 3). Plots show LOG2 fold change and error bars calculated by LOG2 FC SE. **d** Relative mRNA expression for Trp transporters; data extracted from previously published dataset[30], comparing WT and a mouse model of liver carcinoma driven by an inducible Tet-MYC transgene. Plots show RPKM values (*n* = 1). **e**-**i** LC-MS/MS quantification of Trp (**e**), Kyn (**f**), KA (**g**), cinnabarinic acid (CA) (**h**), and serotonin (Ser, **i**) in WT (*n* = 3) and MYC-ON livers (*n* = 3). Plots show mean +/− SEM and *P*-value was calculated by unpaired *t*-test *\*P* ≤ 0.05. **j** Schematic representation of $^{13}$C-Trp infusion in WT and MYC-expressing livers. **k**-**m** Quantification of $^{13}$C-Trp

(**k**), $^{13}$C-Kyn (**l**), and $^{13}$C-KA (**m**) in WT (*N* = 3) and MYC-ON livers (*n* = 3). Plots show mean +/− SEM and *P*-value was calculated by unpaired *t*-test *\*P* ≤ 0.05. **n** NAD$^+$, NADH, and NADP+ measured by LC-MS-MS in WT (*n* = 2) and MYC-expressing livers (*n* = 4) MYC-ON. Plots show mean +/− SEM and *P*-value was calculated by unpaired test *\*P* ≤ 0.05. **o** Quantification of $^{13}$C-NAD$^+$ in WT (*n* = 3), normal looking regions of MYC-overexpressing livers (*n* = 3), and tumors of MYC-overexpressing livers (*n* = 6) after$^{13}$ C-Trp infusion. Plots show mean +/− SEM and *P*-value was calculated by unpaired *t*-test. *\*P* ≤ 0.05. **p** Relative mRNA expression of transporters and enzymes that generate NAD by the salvage pathway plotted from RNA-seq of WT liver (*n* = 2) and liver in which MYC was turned on for 45 days (*n* = 3). Plots show LOG2 fold change, and error bars calculated by LOG2 FC SE. **q** Western blot analysis for SLC1A5, S6, and S6K and Trp-metabolizing enzymes of WT and MYC-expressing livers (*n* = 3).

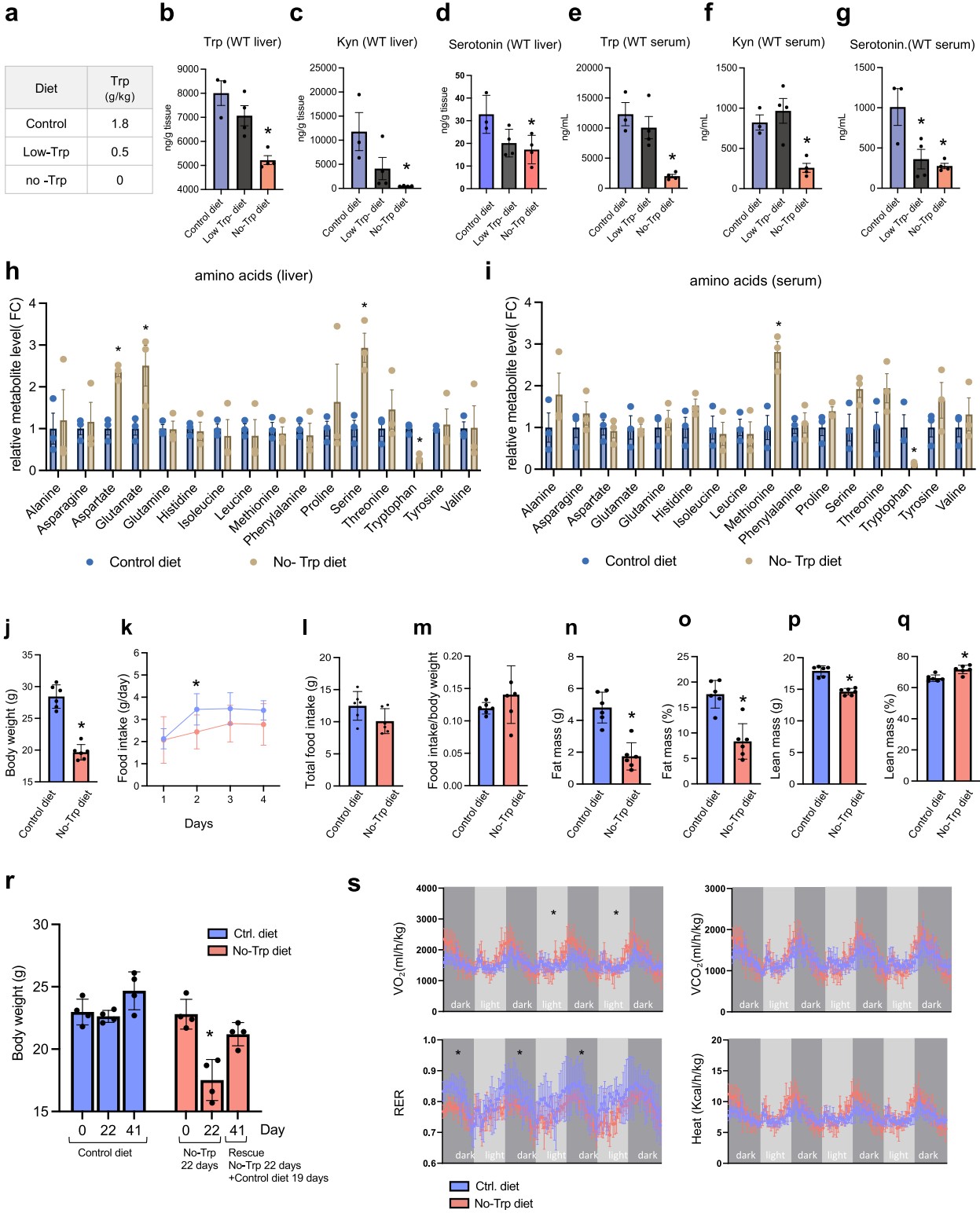

weight (Fig. 3c, d). Moreover, the No-Trp diet did not affect the liver–body ratio in wild type (WT) mice (Fig. 3d), indicating that Trp reduction has a more severe effect on cancer cells in the liver. We confirmed that the levels of Trp and Kyn (Fig. 3e, f) were lower with Trp starvation in the MYC-driven liver tumors.

To compare tumor burden, we fed mice either the control or No-Trp diets for only 21 days and sacrificed all mice at day P50 (Fig. 3g). Consistent with our previous results, mice that were Trp starved had dramatically lower liver weights (Figs. 3h, i, S3a).

Immunohistochemistry (IHC) showed that these livers had fewer and smaller tumors observed by the darker hematoxylin and eosin staining (H&E) (Fig. 3j). The small tumors present in livers of the mice starved of Trp were positive for Ki67 and MYC (Fig. 3j, k), thus demonstrating a reduction in tumor burden, but not absence of tumors. H&E staining showed that Trp-starved tumors did not display gross cellular defects or nucleolar stress (Fig. S3b, e). Although provided a diet lacking Trp prevented liver tumors from forming, more modest effects on extending survival were observed when the No-Trp diet was provided

**Fig. 2 | Effects of Trp depletion in normal mice. a** Trp content in diets. **b–d** Trp (**b**), Kyn (**c**), and serotonin (**d**) in control ($n = 3$), Low-Trp ($n = 4$), and No-Trp diets ($n = 4$). Each dot represents one mouse. Plot shows mean +/− SEM and P-value was calculated by unpaired t-test. *$P \le 0.05$. **e–g** Trp (**e**) Kyn (**f**), and serotonin (**g**) in serum of mice fed the indicated diets for 21 days ($n = 3$). Each dot represents one mouse. Plot shows mean +/− SEM and P-value was calculated by unpaired t-test. *$P \le 0.05$. **h–i** Amino acid in liver (**h**) and serum (**i**) after 21 days in the indicated diets ($n = 3$). Each dot represents one mouse. Plot shows mean +/− SEM and P-value was calculated by unpaired t-test. *$P \le 0.05$. **j** Weight of mice fed control or No-Trp diet ($n = 6$). The plots show mean with SEM and the P-value was calculated by unpaired t-test. *$P \le 0.05$. **k–m** Daily food consumption (**k**), cumulative (**l**), and per weight (**m**) in control or No-Trp diets day 18–21 ($n = 6$). The plots show mean with SEM and the P-value was calculated by unpaired t-test *$P \le 0.05$. **n–o** Fat mass (**n**) and percentage (**o**) measured by ECO MRI after 21 days on the diets ($n = 6$). The plots show mean with SEM and the P-value was calculated by unpaired t-test. *$P \le 0.05$. **p–q** Lean mass (**p**) and percentage (**q**) measured by ECO MRI after 21 days in the diets ($n = 6$ mice). The plots show mean with SEM and the P-value was calculated by unpaired t-test. *$P \le 0.05$. **r** Weight of animals fed No-Trp and re-fed complete diet ($n = 4$). Each dot represents one mouse. Plot shows mean +/− SEM and P-value was calculated by unpaired t-test. *$P \le 0.05$. **s** $O_2$ consumption ($VO_2$), $CO_2$ production ($VCO_2$), respiration rate (RER) and heat production (Heat) in mice fed the control or No-Trp diet on day 18-21 ($n = 4$). Plot shows mean +/− SEM. Area under the curve with mean with SD and the P-value calculated by unpaired t-test. *$P \le 0.05$.

to mice with advanced disease (Fig. 3l–m). Moreover, when Trp was re-introduced to the diet of Trp-starved MYC-ON mice, their tumors grew, but this growth was delayed (Fig. 3n–o). These data suggest that brief Trp starvation halted tumor growth but did not permanently impair it.

## Trp starvation rescues a normal liver expression profile in MYC-ON tumors

To define the molecular changes induced by Trp starvation in liver tumors, we performed RNA-seq of livers with tumors driven by MYC. The mice were fed either the control or No-Trp diet for 21 days prior to harvesting RNA. We then compared the pattern of genes regulated by Trp starvation with the genes regulated by the activation of MYC in the liver (Fig. 4a–d). We found an overlap in the gene ontology (GO) of genes upregulated by MYC (Fig. 4a) and downregulated by Trp starvation (Fig. 4d) including cell cycle and DNA replication. Moreover, there was also an overlap between genes downregulated by the activation of MYC and genes upregulated by Trp starvation (Fig. 4b, c). 1147 genes upregulated by MYC were downregulated upon Trp starvation and those genes were mostly related to cell growth (Fig. 4e). On the other hand, 1333 genes downregulated by activation of MYC were upregulated by Trp starvation and those genes were mostly related to lipid metabolism, a function performed by normal liver cells (Fig. 4f). A heatmap gated for the changes induced by Trp starvation confirm that starving MYC-ON livers of Trp rescues a transcriptional signature that resembles the gene signature present in WT livers (Fig. 4g). We confirmed that genes involved in cell cycle, DNA replication, and ribosome biogenesis were among the pathways significantly downregulated (Fig. 4h–j). Expression was repressed upon Trp starvation for 25 genes involved in DNA replication machinery (Fig. 4h), 32 genes involved in multiple steps of ribosome biogenesis (Fig. 4i), and 8 cyclin genes (Fig. 4j). Expression was upregulated by Trp starvation for Ccnl2 and Ccni (Fig. 4j). RNA polymerase sub-units were also downregulated (Fig. 4k). On the basis of the transcripts downregulated upon Trp starvation and the motifs in their promoters, we inferred that the activities of transcription factors are likely lower with Trp starvation. Interestingly, we found that the most downregulated gene signature was downstream of MYC (Fig. 4l). The expression of MYC in the livers from mice fed the No-Trp diet was also lower than in livers from mice fed the control diet (Fig. 4m). Nevertheless, MYC was still expressed at significantly higher levels in the MYC-ON Trp-starved livers than in the WT livers (Fig. 4m), suggesting that there may be additional mechanism driving cell growth via the utilization of Trp in liver tumors.

## Mouse and human liver cancer cell xenografts are sensitive to Trp starvation

Using RNA-seq from the TCGA data set for liver cancer, we found that about 40% of patients with HCC had elevated SLC7A5, 50% had elevated SLC1A5 (Fig. 5a) and 64% had either or both elevated. IDO1 was elevated in 50% of liver cancer samples while IDO2, TDO2, and AFMID were predominantly downregulated (Fig. 5a). Expression of SLC1A5, but not IDO1, TDO2, KYNU, and AFMID, was strongly associated with

shortened survival of HCC patients (Fig. S4a). Trp levels were higher in the tumor than in morphologically normal liver tissue from the same patient (Fig. 5b); Kyn, KA, and serotonin were not significantly altered (Fig. 5c–e). Analysis of samples from individual patients with HCC revealed that 7 of the 10 patients had higher Trp levels in their tumor than in their benign tissue samples, and only 1 patient had a higher Kyn:Trp ratio in their tumor than in benign tissue samples (Fig. S4b, d). In summary, human liver tumors have variable levels of Kyn but higher Trp levels.

Human HCC cell lines HUH7, SNU449, and HepG2 expressed elevated levels of SLC1A5 and SLC7A5 in comparison to normal THLE2 liver cells (Fig. S4e). Silencing of MYC, SLC1A5, and SLC7A5 significantly stunted in vitro growth of HUH7 cells (Fig. 5f). Moreover, knocking MYC down by siRNA in HUH7 resulted in significantly less of the expression of SLC1A5 and SLC7A5 (Fig. 5g), thus indicating that MYC is necessary for the expression of these transporters in human cells. To determine whether tumors arising from different genetic alterations were also dependent on higher Trp levels and were sensitive to Trp starvation, we generated a primary mouse liver cancer cell line HCC53N[34] from tumors driven by the combination of p53 KO and the overexpression of mutant N-Ras both found to occur in HCC[35,36]. Like HUH7, knocking down MYC, SLC1A5, and SLC7A5 in these cells resulted in less cell proliferation (Fig. 5h) and the silencing of MYC resulted in less expression of SLC1A5 and SLC7A5 (Fig. 5i).

We characterized the time for in vivo growth of HCC53N and HUH7 cells as xenografts in NOD SCID mice (Fig. S4f) and use these 2 cell lines to study the importance of Trp for their growth. Both cell lines express higher levels of MYC than WT liver, thus supporting the importance of MYC in their growth (Fig. 5j). Trp starvation limited the growth of HUH7 cells xenotransplanted into NOD SCID mice (Figs. 5k, S4g). Similar growth limitations were seen in HCC53N cells xenotransplanted (Figs. 5l, S4h). Trp starvation also resulted in lower MYC levels in xenotransplanted HUH7 cells (Fig. 5m) but not in HCC53N (Fig. 5n), suggesting that Trp starvation limits tumor growth by mechanisms that do not require MYC repression. Our results demonstrate that depletion of dietary Trp is efficient at limiting the growth of liver tumors arising from different genetic mutations and backgrounds. Trp starvation caused modest weigh loss in NOD SCID mice (Fig. S4i).

## Trp starvation does not impair protein synthesis in liver cancer cells

Given that Trp is an essential amino acid, we reasoned that elimination of Trp from the diet leads to a reduction in protein synthesis, thus limiting cell growth in vivo and in vitro. Indeed, incorporation of Trp into proteins is enhanced when MYC is turned on in the liver (Fig. 6a). These observations led us to compare protein synthesis in liver cancer cells grown in vitro and in vivo in the presence or absence of Trp. To determine whether Trp starvation in vivo affected protein synthesis in liver tumors, we fed animals with control diet or No-Trp diet for 3 weeks and infused these mice with $^{13}C$-glutamine. We then measured the incorporation of $^{13}C$-glutamine into proteins in the livers by mass

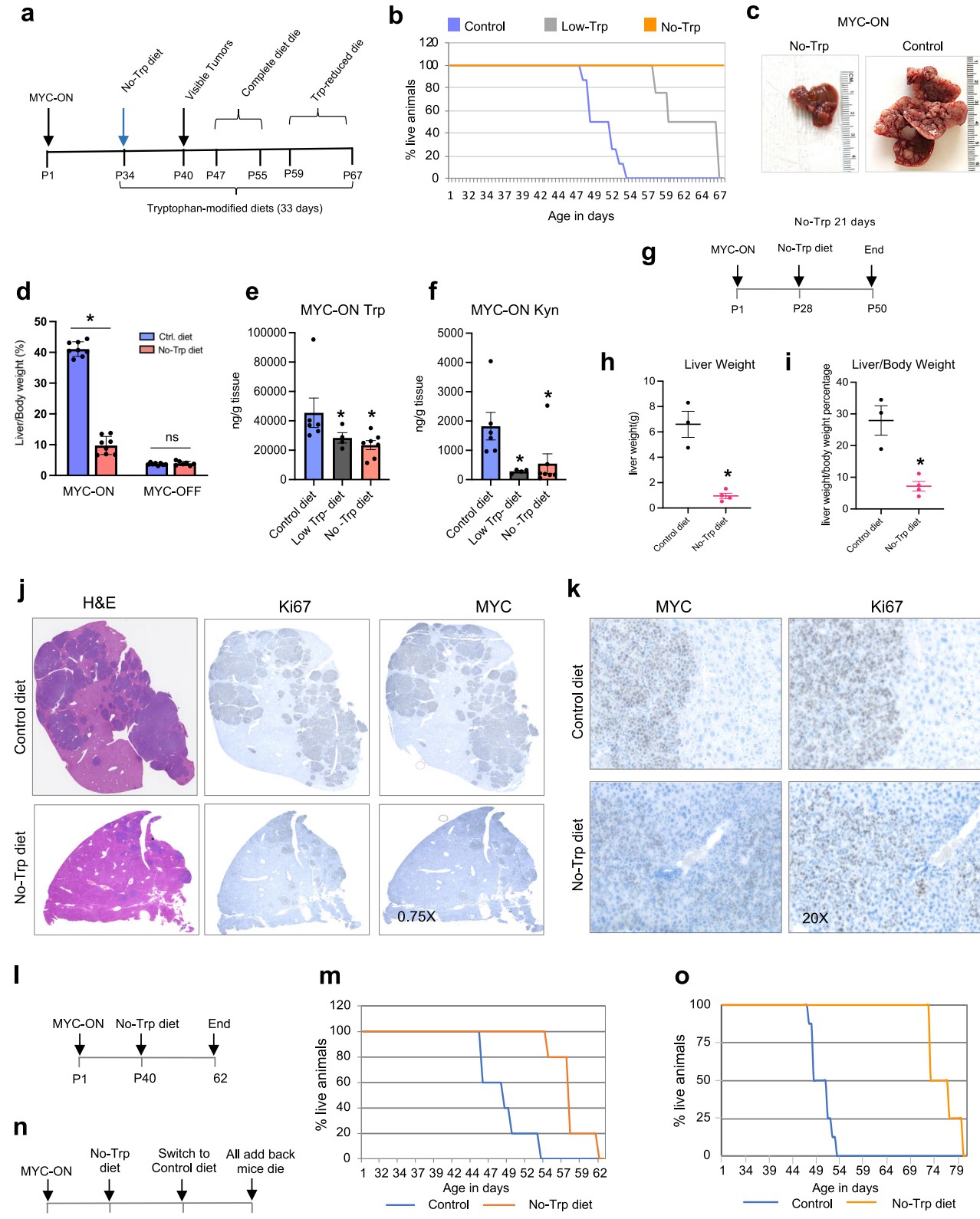

spectrometry 3 h after infusion. No difference in incorporation of [13]C-glutamine was demonstrated between the livers from mice fed either the control or No-Trp diet (Fig. 6b, c).

To directly measure protein synthesis upon short-term Trp starvation, we performed puromycylation in HUH7 and HCC53N and found that puromycyn incorporation had no reduction between newly synthesized peptides without Trp (Figs. 6d, S5a, b). We also measured translation by puromycylation using culture media prepared with dialyzed serum to ensure that the Trp present in the serum was not compensating for the lack of Trp in the media. We found that even using dialyzed serum, Trp starvation did not ablate translation rates in liver cancer cells and in some cases high levels of Trp-reduced translation (Fig. S5c–e). Overnight starvation of Trp prior to stimulation with increasing amounts of Trp also did not affect the overall

**Fig. 3 | Trp deprivation reduces the growth of MYC-driven liver tumors.**
**a** Survival study. **b** Survival curve of mice assigned in control, Low-Trp, or No-Trp diets ($n = 8$ mice per each group). **c** Liver of animals assigned to control and No-Trp diet in (**b**). **d** Ratio of liver to body weight of MYC-ON and WT mice fed either the control or No-Trp diet. MYC-ON control diet ($n = 8$), MYC-ON No-Trp diet ($n = 8$), WT control ($n = 8$), WT No-Trp diet ($n = 7$). Each dot represents one mouse. Plot shows mean +/− SEM and $P$-value was calculated by unpaired $t$-test. *$P \leq 0.05$.
**e**–**f** Trp and Kyn in the MYC-ON mice fed the control, Low-Trp, or No-Trp diets at day 67 of age ($n = 6$ mice per group). Each dot represents one mouse. Plot shows mean +/− SEM and $P$-value was calculated by unpaired $t$-test. *$P \leq 0.05$. **g** MYC-ON mice were assigned to the control diet or No-Trp diet for 21 days. **h** Weight of the liver of MYC-ON mice fed either the control ($n = 3$) or No-Trp diet ($n = 4$) for 21 days. SEM and $P$-value was calculated by unpaired $t$-test. *$P \leq 0.05$. **i** Ratio of liver weight

to body weight of MYC-ON mice after being fed either the control ($n = 3$) or No-Trp diet ($n = 4$) for 21 days. SEM and $P$-value was calculated by unpaired $t$-test. *$P \leq 0.05$. **j**–**k** Representative H & E, Ki-67, and MYC staining of livers obtained from mice fed either the control or No-Trp diet for 21 days (as described in **g**) at 0.75X (**j**) or 20X (**k**) magnification. **l** Schematic of the survival study. MYC-ON mice were randomly assigned to either the control or No-Trp diet, and survival of mice was determined. **m** Survival curve denoting mice assigned to the control ($n = 8$) and the No-Trp diet ($n = 4$). **n** Schematic of the survival study. MYC-ON mice were assigned to the No-Trp diet and changed back to the control diet after 21 days, and survival of mice was measured. **o** Survival curve of mice that were fed for 21 days on No-Trp diet and switched back to the control diet ($n = 4$). The survival was compared animals fed control diet ($n = 8$).

translation of liver cancer cells (Fig. S5f, g). To increase the rigor of these experiments, we measured translation initiation via Click-IT chemistry with L-azidohomoalanine (AHA), which provides a fast, sensitive, non-radioactive technique to measure protein synthesis in liver cancer cells. We found that Trp elimination had no effect on overall translation in cultured cells (Fig. 6e). Moreover, analyzing the results of mass spectrometry of the proteins expressed in MYC-ON livers in mice fed either the control or No-Trp diets for 3 weeks, we found no difference in the abundance of Trp in the up- or down-regulated proteins, suggesting that the Trp content in expressed proteins is not affected by Trp starvation (Fig. 6f). Trp starvation was previously proposed to increase translational errors, leading to the incorporation of Phe, Tyr, Ile or Leu instead of Trp for the Trp codon in cultured cells[37,38]. Importantly, Trp starvation did not cause observable alterations in Trp>Phe/Tyr/Ile/Leu substitutions in the livers of MYC-ON mice or xenografted HCC53N cells (Fig. 6g, h). We found 91 peptides with Trp substitutions in the MYC-ON livers of mice fed the control diet and 92 substitutions in livers of mice fed the No-Trp diet. The identity of these peptides varies, but the frequency of substitution at the Trp codon does not. We concluded that Trp starvation does not cause amino acid substitutions in liver tumors.

The sustained protein synthesis found in Trp-starved cells can be explained by the low incidence of Trp in proteins. Importantly, this can also be explained by the increased expression of mRNAs encoding for the SLC family of transporters in livers of MYC-ON mice starved of Trp (Fig. S5j, k). Among these the expression levels of several amino acid transporters were also altered (Fig. S5k), suggesting an increase in nutrient transport activity into Trp-free cells. Strikingly, the expression of the Trp transporter, SLC1A5, was dramatically higher in cells and MYC-ON livers starved of Trp (Fig. 6I, j). Markers of autophagy and ATF4 were not affected (Fig. S5h, i). We surmised that cells experiencing Trp starvation increase the expression of cellular transporters including the Trp transporter SLC1A5 to maintain a rate of protein synthesis similar to the rate of the cells with Trp.

### I3P rescues the growth of Trp-starved cells in vitro and in vivo
Given that Trp deprivation had no significant effect on protein synthesis within our experiments, we asked whether the loss of Trp-derived metabolites is responsible for the growth restriction of liver tumors and cells in vitro and in vivo. We examined the ability of the first downstream metabolite in each of the Trp metabolism pathways for their ability to rescue the growth of Trp-starved liver cancer cells: these are Kyn, 5HTP (a cell permeable intermediate of serotonin) and I3P, which is the first indole generated by IL4I1 (Fig. 7a). Cell viability was reduced by 50% for HUH7 cells cultured in medium lacking Trp compared to HUH7 cells cultured with Trp medium. The addition of Kyn, nicotinic acid (NA) or nicotinamide (NMN) to the culture media, which reconstituted the pool of NAD⁺ by the salvage pathway, did not rescue the cells (Fig. 7b). Only I3P was capable of partially rescuing the growth of HUH7 cells (Fig. 7c). Increasing amounts of I3P led to further increases in the growth of Trp-starved HUH7 cells (Fig. 7d).

Importantly, I3P did not increase the growth of Trp-fed cells, indicating that the growth advantage provided by I3P is the active metabolite downstream of Trp (Fig. 7e). Moreover, the I3P downstream product indole-3-aldehyde (I3A), indole-3-lactic acid (ILA) had no effect on the growth of Trp-starved cells (Fig. 7f). Like HUH7 cells, the growth of HCC53N cells was partially rescued by I3P (Fig. 7g, h). No growth advantages were observed in Trp-fed HCC53N cells (Fig. 7i). Supplementation with products downstream of I3P had no effect on growth (Fig. 7j). We confirmed that I3P can be taken up by HCC53N cells grown in complete media in vitro (Fig. 7k) and that overnight starvation of Trp in HCC53N cells is sufficient to cause a reduction in Trp, I3P and its downstream metabolites (Fig. 7l). Strikingly, I3P is present at very high levels, much higher levels than Trp itself, in liver cancer cells, an indication that this metabolite likely plays a major role in the biology of these cells.

To determine whether I3P can rescue tumor growth in vivo, we transplanted HCC53N cells into NOD SCID mice fed a No-Trp diet and treated these mice daily with intra peritoneal (IP) injections of either vehicle or I3P (Fig. 7m). I3P was biologically active in vivo and drove the growth of Trp-starved HCC53N cells (Fig. 7n, o) but I3P treatment had no effect on the body weight of Trp-starved mice (Fig. 7p,). Importantly, these xenografts demonstrated an increase in I3P when animals were fed a No-Trp diet supplemented with I3P (Fig. 7q). However, the levels of products downstream of I3P were modestly affected (Fig. 7q). Other Trp metabolites in the Kyn pathway were unaffected by I3P supplementation (Fig. S6).

### I3P levels are higher in MYC-ON liver tumors and rescue the growth of Trp-starved tumors in vivo
Although the metabolites in the Kyn pathway were lower with MYC activation in the liver (Fig. 1), the I3P level was much higher than in the WT mice (Fig. 8a). The levels of the products downstream of I3P were not altered. Like the liver cancer cell lines, liver tumors display high levels of I3P, an indication of its biological relevance (Fig. 8a) Importantly, Trp starvation causes a reduction of I3P in the liver (Fig. 8b), thus strengthening the possibility that I3P is a potential mediator of the oncogenic function of Trp. In agreement with elevated levels of I3P, the enzyme IL4I1, which generates I3P from Trp, was also higher in MYC-ON tumors (Fig. 8c). Expression of IL4I1 correlates negatively with survival of patients with HCC (Fig. 8d) and IL4I1 expression is higher human HCC tumor samples in comparison with normal samples deposited in the TCGA (Fig. 8e). The increase in IL4I1 in liver tumors compared with other Trp-metabolizing enzymes is also in agreement with our results (Fig. S7).

To determine whether I3P rescues tumor growth of MYC-ON Trp-starved mice, we supplemented MYC-ON mice fed the No-Trp diet (Fig. 8f) with daily IP injections of either vehicle or I3P. I3P potently rescued the growth of liver tumors as shown by liver morphology (Fig. 8g) and liver weight (Fig. 8h) without affecting the overall weight of the mice (Fig. 8i). We confirmed that I3P levels, but not Trp, were elevated in the livers of MYC-ON mice fed a No-Trp diet supplemented

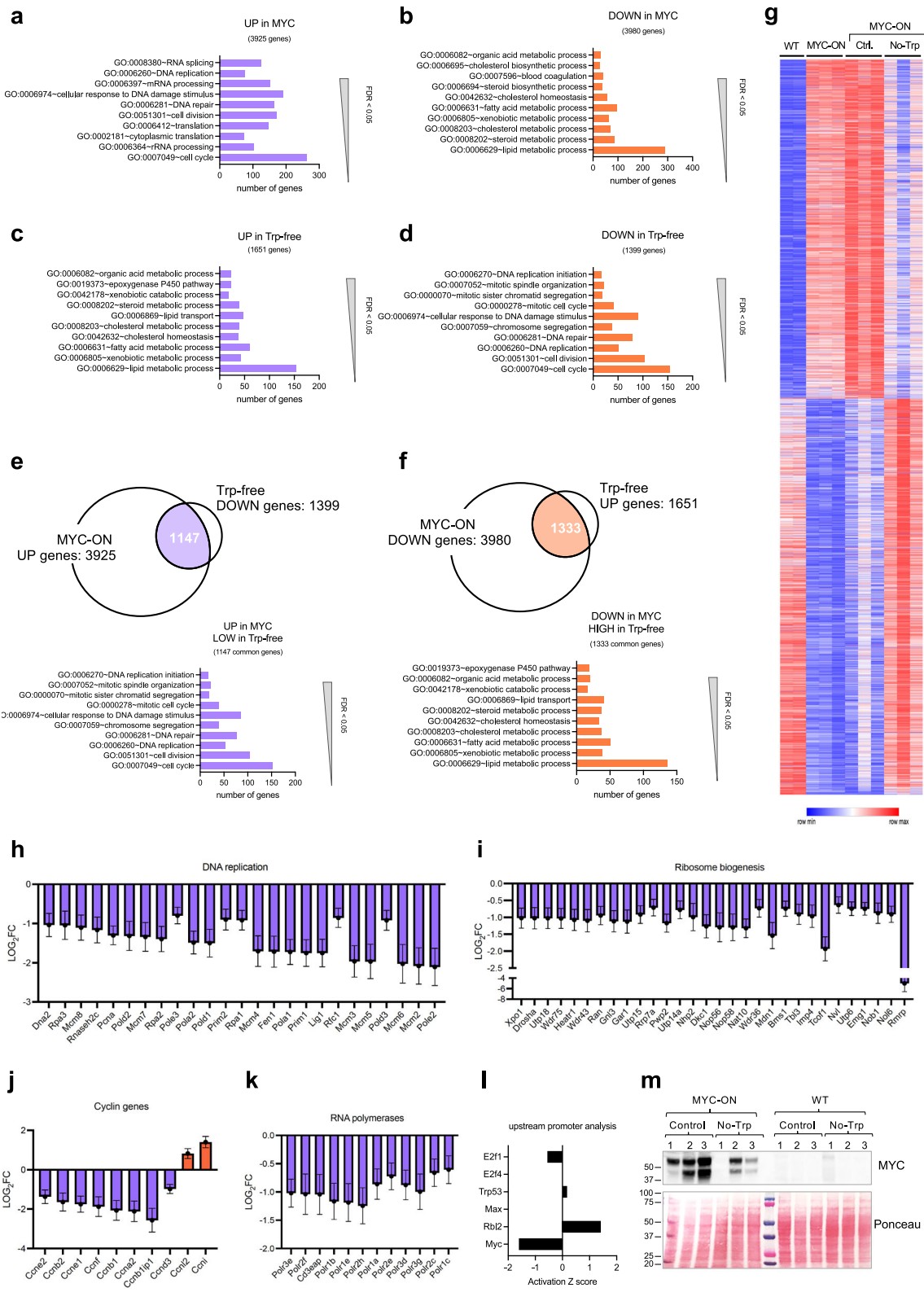

with I3P (Fig. 8j). As opposed to the results obtained using xenografts, Kyn and the serotonin precursor, 5HIAA, were reduced in the livers of MYC-ON mice fed No-Trp diet supplemented with I3P (Fig. S8a). However, it is not likely that this decrease plays a role on the oncogenic functions of I3P. Metabolites altered in the liver were also altered in circulation (Fig. S6b). Importantly incubation I3P supplementation in cells grown in media with or without Trp had no effects on cellular Trp levels (Fig. 8k). These results indicate that I3P supplementation in vitro and in vitro cannot reconstitute Trp levels.

I3P was previously shown to perform growth-promoting functions via the ability to bind to and drive the nuclear translocation of the transcription factor AHR in glioblastoma cells[27]. Moreover, we found that silencing AHR in HCC53N cells partially reduced their growth in the No-Trp media supplemented with I3P (Fig. 8l). To determine

**Fig. 4 | Trp deprivation rescues normal transcriptional programs in MYC-ON livers. a** Gene ontology (GO) of genes upregulated by the expression of MYC in the liver measured by RNA-seq comparing WT livers (*n* = 2) and livers that had MYC-ON for 45 days (*n* = 3). **b** GO of genes downregulated by the expression of MYC in the liver measured by RNA-seq comparing WT livers (*n* = 2) and livers that had MYC-ON for 45 days (*n* = 3). **c** GO analysis of the genes upregulated in the liver upon Trp starvation in MYC-ON mice fed either the control or No-Trp diet for 21 days (*n* = 3 mice per sample), showing the top 10 significant pathways. **d** GO analysis of the genes downregulated in the liver upon Trp starvation in MYC-ON mice fed either the control or No-Trp diet for 21 days (*n* = 3 mice per sample), showing the top 10 significant pathways. **e** Venn diagram showing the overlap and GO of genes upregulated by MYC and downregulated by 21-day Trp starvation in the liver of mice. GO analysis of the genes upregulated upon Trp starvation in MYC-ON livers,

showing the top 10 significant pathways affected. **f** Venn diagram showing the overlap and GO of genes downregulated by MYC and unregulated by 21-day Trp starvation in the liver of mice. GO analysis of the genes upregulated upon Trp starvation in MYC-ON livers, showing the top 10 significant pathways affected. **g** Heatmap of genes regulated by Trp starvation and compared with WT, and MYC-ON liver transcriptional signatures. **h–k** Relative mRNA levels of genes involved DNA replication (**h**) ribosome biogenesis (**i**), cyclin genes (**j**), and RNA polymerase genes (**k**) that are differentially expressed between the control (*n* = 3 mice) and No-Trp diet (*n* = 3 mice) in MYC-ON livers. Plots show LOG2 fold change, and error bars calculated by LOG2 FC SE. **l** Upstream promoter analysis of the genes regulated by Trp starvation for 21 days the livers of MYC-ON mice. **m** MYC transgene measured in livers of mice after Trp starvation for 21 days (*n* = 3 mice for each group shown).

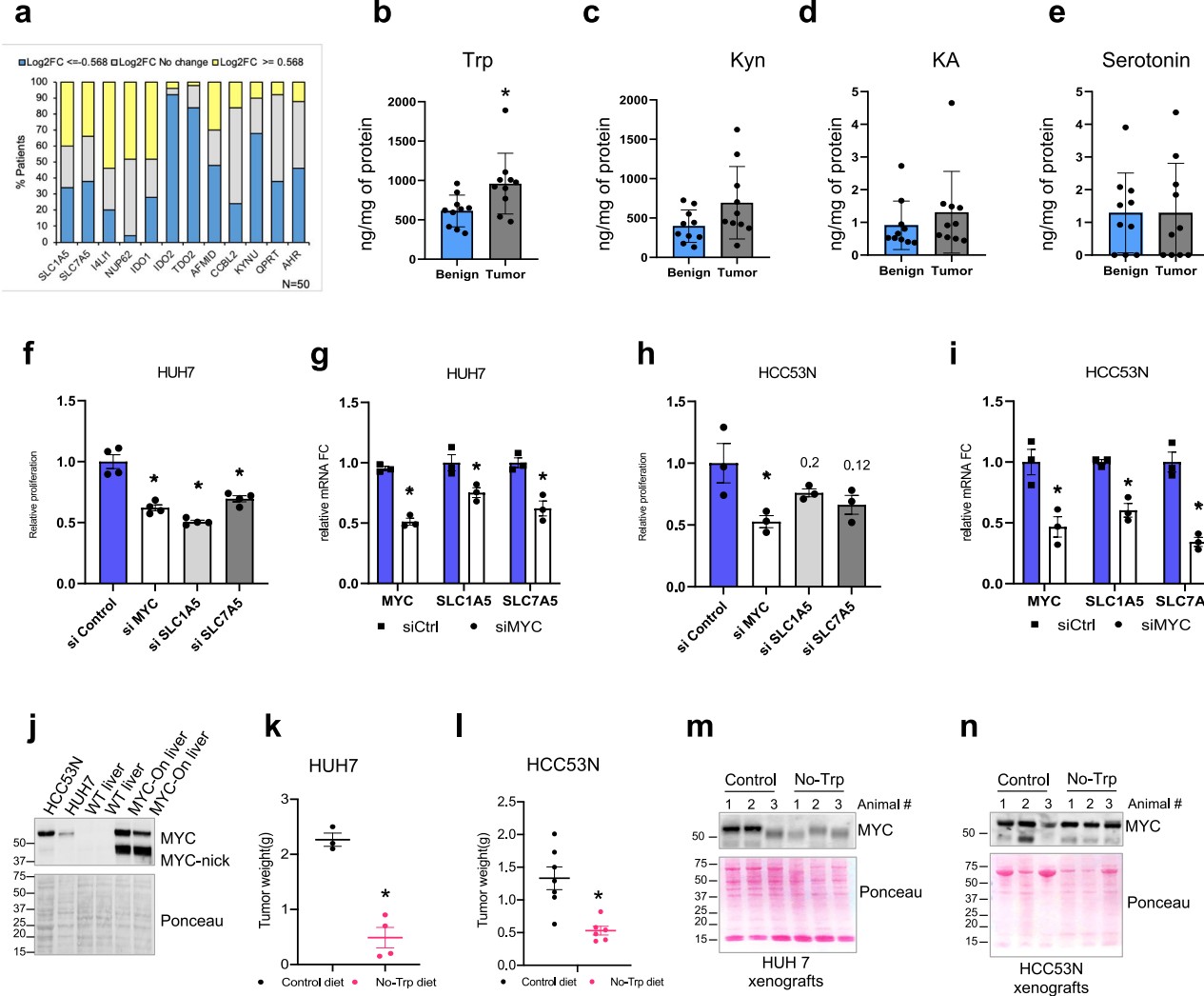

**Fig. 5 | Mouse and human liver cancer cells are sensitive to Trp starvation.**
**a** Expression of indicated genes in hepatocellular carcinoma (HCC) patients comparing normal and tumor tissues from the same patient. Upregulation ≥ 1.3X. Downregulation ≤ 0.7X. No change is between 0.7X and 1.3X. **b–e** Trp (**b**) Kyn (**c**) and KA (**d**) and serotonin (**e**) in benign tissue or tumor from HCC patients (*n* = 10). Each plot represents one patient sample. Plot shows mean +/− SEM and *P*-value was calculated by unpaired *t*-test. *\*P* ≤ 0.05. **f** Viability of HUH7 cells transfected with control, MYC, SLC1A5, and SLC7A5 siRNA measured by crystal violet, (*n* = 4). Plot shows mean +/− SEM and *P*-value was calculated by unpaired *t*-test. *\*P* ≤ 0.05. **g** RT-qPCR for MYC, SLC1A5, and SLC7A5 in HUH7 cells transfected with control or MYC siRNA. mRNA levels are shown as relative fold change to control siRNA and (*n* = 3). Plot shows mean +/− SEM and *P*-value was calculated by unpaired *t*-test. *\*P* ≤ 0.05.

**h** Viability of HCC53N cells 3 days after transfection of control or MYC, SLC1A5, and SLC7A5 siRNA measured by crystal violet (*n* = 3 per sample). Plot shows mean +/−SEM and *P*-value was calculated by unpaired *t*-test. *\*P* ≤ 0.05. **i** RT-qPCR for expression of MYC, SLC1A5, and SLC7A5 mRNA in HCC53N cells transfected with control or MYC siRNA (*n* = 3). mRNA levels are shown as relative fold change to control siRNA. Plot shows mean +/− SEM and *P*-value was calculated by unpaired *t*-test. *\*P* ≤ 0.05. **j** Western blot for MYC in HCC53N and HUH7 cells compared to WT or MYC-ON liver, repeated 3X. **k–l** Tumors of xenografs of HUH7 (**k**) and (HCC53N) (**l**) of cells grown in mice fed either the control (*n* = 3) or No-Trp diet for 21 days (*n* = 4). Plot shows mean +/− SEM and *P*-value was calculated by unpaired *t*-test. *\*P* ≤ 0.05. **m, n** Western blot of lysates from HUH7 (**m**) of HCC53N (**n**) xenografts grown in NOD SCID mice fed either the control or No-Trp diet for 21 days (*n* = 3).

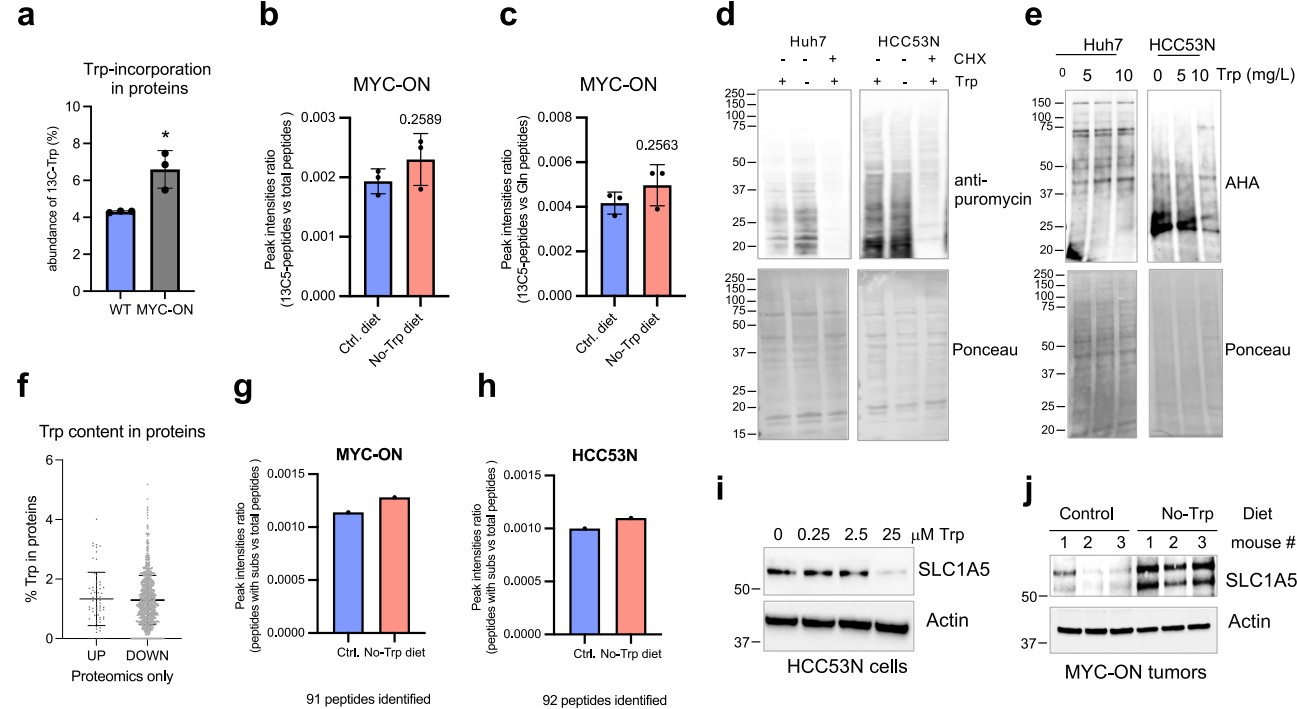

**Fig. 6 | Trp deprivation does not affect protein synthesis in liver cancer.**
**a** Incorporation of $^{13}$C-Trp into peptides in WT and MYC-ON livers ($n = 3$). Each dot represents one mouse. Plots show mean +/− SD and $P$-value was calculated by $t$-test. **b** Peak intensities of peptides containing $^{13}$C-glutamine normalized by total amount of peptides detected by proteomics in livers from mice fed either the control or No-Trp diet 3 h after tail vein injection of $^{13}$C-glutamine ($n = 3$). Each dot represents one mouse. Plots show mean +/− SD and $P$-value was calculated by $t$-test. **c** Peak intensities of peptides containing $^{13}$C-glutamine normalized by all glutamine-containing peptides detected by proteomics in livers from mice fed either the control or No-Trp e diet. $P$-value indicated above bar. ($n = 3$) Each dot represents one mouse. Plots show mean +/− SD and $P$-value was calculated by $t$-test. **d** Protein synthesis measured in cells grown in the presence or absence of Trp. Puromycyn with or without cycloheximide (CHX) was added during the final 1 h of the experiment, repeated 3X. **e** Protein synthesis measured by Click-IT™ AHA in cells grown in the absence or presence of Trp, repeated 3X. **f** Percentage of Trp in proteins whose abundance increased (UP) or decreased (DOWN) upon Trp starvation in MYC-ON livers. Each dot represents one protein. Plots show mean +/− SD and $P$-value was calculated by $t$-test. **g** Peak intensities of peptides containing Trp substituted by tyrosine, phenylalanine, leucine, or isoleucine normalized by total peptides detected by proteomics in livers from MYC-ON mice fed either the control or No-Trp diet ($n = 3$). Each dot represents the ratio using the average of 3 replicates. **h** Peak intensities of peptides containing Trp substituted by tyrosine, phenylalanine, leucine, or isoleucine normalized by total peptides detected by proteomics in HCC53N xenografts from mice fed the control or No-Trp diet ($n = 3$). Each dot represents the ratio using the average of 3 replicates. **i** Western blot of HCC53N lysates grown overnight in the indicated media, repeated 3X. **j** Western blot of MYC-ON liver lysates from mice fed either the control ($n = 3$) or No-Trp ($n = 3$). *$P \le 0.05$.

whether I3P had the ability to promote AHR translocation in liver cancer cells, we measured the nuclear and cytosolic AHR fractions in HCC53N cells 1 h after incubation with I3P (Fig. 8m) and found that I3P efficiently drove nuclear translocation of AHR but had no effect of MYC levels or localization. Longer incubations with I3P confirmed that I3P promotes the expression of the canonical AHR targets CYP1A1, NQO1, AHRR, and to a lower extent SCIN and UMPS (Fig. 8n), suggesting that AHR activation may be involved in mediating I3P-induced growth. However, I3P had no effect on MYC or its partner MAX (Fig. 8n). Moreover, we found that protein synthesis (Fig. S9a) or MYC levels (Figs. 8m, n, S9b) were not consistently affected in cells or tumors supplemented with I3P, indicating that these do not mediate cancer cell growth induced by I3P. We, therefore, concluded that I3P promotes cell growth by mechanisms that are independent of MYC activation that may require AHR-dependent pathways Fig. 9.

## Discussion

Excessive alcohol consumption, viral hepatitis, metabolic syndrome, and obesity are all major causes of chronic inflammation and liver damage. The culmination of these insults dramatically increases the risk of HCC[8,39]. Likely due to these environmental factors, the incidence of HCC has tripled in recent decades. However, the overall survival rate of patients with HCC has remained poor[40]. These population-wide factors necessitate the discovery of new treatments for HCC. We

propose that a better understanding of the nutritional needs of liver cancer cells may open avenues for novel therapeutic interventions.

We demonstrated that MYC-driven liver tumors display enhanced Trp uptake compared to normal livers yet downregulate metabolism along the Kyn pathway. This is opposite to many cancers such as cancers of the colon, pancreas, breast, and brain in which non-cancerous tissue displays a significantly lower expression of Kyn pathway enzymes than tumor tissue[13,24,41–45]. Furthermore, we found that MYC-driven liver oncogenesis requires Trp (Fig. 9). When MYC-ON mice were starved of Trp, their livers retained a normal phenotype with a transcriptome more akin to the normal liver than tumor.

Although the activity of most Trp-metabolizing enzymes in the Kyn pathway is less in HCC, we found that the Trp-metabolizing enzyme IL4I1 is higher in human and mouse liver tumors than in noncancerous tissues. Furthermore, we identified I3P, the enzymatic product of IL4I1, to be the sole Trp catabolite capable of rescuing the growth of Trp-starved liver tumors in vitro and in vivo. Interestingly, MYC-driven tumors specifically accumulated I3P as opposed to its catabolites. This finding is seemingly in opposition to the findings of previous studies, which found that IL4I1-expressing glioblastoma cells had undetectable I3P levels and high levels I3A, ILA, and KA[27].

We found that I3P is a ligand for AHR in HCC cells and can drive its nuclear translocation. Others have found that ablation of AHR

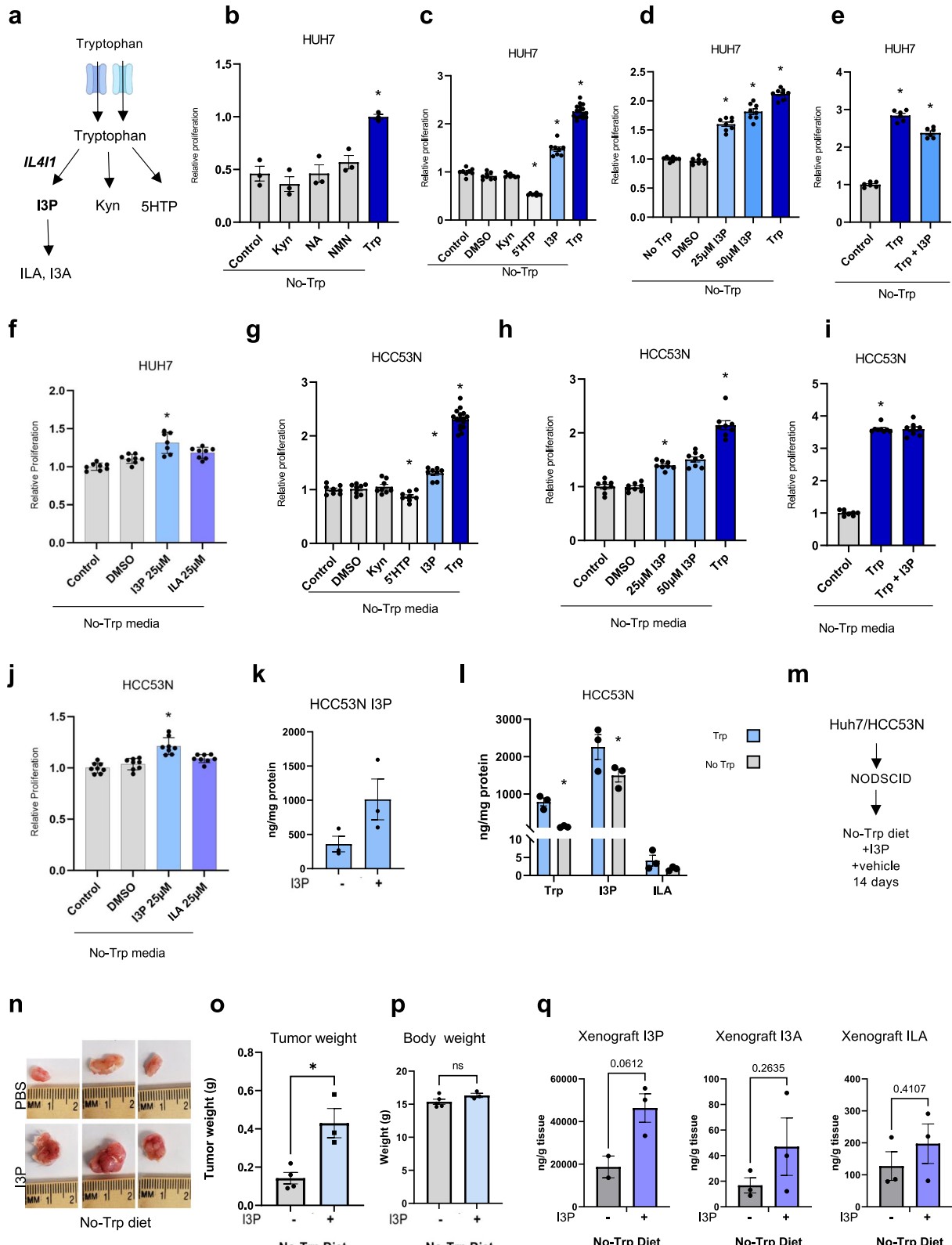

suppresses cell growth, whereas overexpression promotes cell proliferation and enhances tumorigenic activity in vitro[46]. Constitutive activation of AHR promotes liver cancer in vivo[47]. However, the specific link between I3P, AHR, and HCC has not yet been explored. IL4I1, its metabolic pathway, and its role in tumorigenesis are relatively new considerations that require additional research. It is possible that the developing "I3P pathway" like the Kyn pathway has a nuanced role in

cancer with distinct activities not solely dependent on AHR. Our results demonstrate a major cell autonomous component in limiting tumor growth by Trp starvation and in the pro-tumoral role of IL4I1 in HCC. However, future studies are necessary to determine how long-term administration of a reduced Trp diet may affect the tumor microenvironment. It is possible that reducing I3P levels may promote an enhanced immune response.

**Fig. 7 | I3P increases the growth of Trp-starved tumors. a** Trp metabolism. ILA: Indole-3-lactic acid, I3A: Indole-3-carboxaldehyde. **b** HUH7 in No-Trp with Trp (75 μM), Kyn (20 μM), nicotinic acid (NA) (25 μM), or nicotinamide (NMN) (25 μM) (*n* = 3). Plot shows mean +/− SEM and *P*-value calculated by one-way ANOVA. *$P \le 0.05$. **c** Viability of HUH7 in No-Trp with 25 μM of Trp (*n* = 16), Kyn (*n* = 8), 5HTP (*n* = 8), or I3P (*n* = 8). Plot shows mean +/− SEM and *P*-value calculated by one-way ANOVA. *$P \le 0.05$. **d** HUH7 in No-Trp with Trp (*n* = 8), or I3P (*n* = 8). Plot shows mean +/− SEM and *P*-value calculated by one-way ANOVA. *$P \le 0.05$. **e** HUH7 with Trp (*n* = 8) with and without I3P (*n* = 8). Plot shows mean +/− SEM and *P*-value calculated by one-way ANOVA. *$P \le 0.05$. **f** HUH7 in No-Trp with DMSO (*n* = 8), I3P (*n* = 7), or ILA (*n* = 8). Plots show mean +/− SEM and *P*-value calculated by one-way ANOVA. *$P \le 0.05$. **g** HCC53N in No-Trp with Trp (75 μM (*n* = 16), Kyn (20 μM) (*n* = 8), 5HTP (25 μM) (*n* = 8), or I3P (25 μM) (*n* = 8). Plot shows mean +/− SEM and *P*-value calculated by one-way ANOVA. *$P \le 0.05$. **h** HCC53N in No-Trp with Trp or I3P (*n* = 8). Plot shows mean +/− SEM and *P*-value calculated by one-way ANOVA.

*$P \le 0.05$. **i** HCC53N in Trp (25 μM) with and without I3P (25 μM) (*n* = 8). Plot shows mean +/− SEM and *P*-value calculated by one-way ANOVA. *$P \le 0.05$. **j** HCC53N in No-Trp with DMSO, I3P, or ILA (*n* = 8). Plots show mean +/− SEM and *P*-value calculated by one-way ANOVA. (*) $P \le 0.05$. **k** I3P in HCC53N in the absence or presence of I3P (25 μM) (*n* = 3) for 1 h. Plot shows mean +/− SEM and *P*-value calculated by one-way ANOVA. *$P \le 0.05$. **l** Trp, I3P, and ILA in HCC53N cultured for 24 h in control or No-Trp (*n* = 3). Plot shows mean +/− SEM and *P*-value calculated by two-way ANOVA. *$P \le 0.05$. **m** Xenografts. **n** HCC53N xenografts images in the No-Trp diet plus daily injections of vehicle or I3P (75 μM). **o–p** Weight of HCC53N xenografts (**o**) and of NOD SCID (**p**) mice fed the No-Trp diet plus vehicle (*n* = 4) or I3P injections (*n* = 3). Each dot represents one mouse. Plots show mean +/− SEM and *P*-value calculated by *t*-test. *$P \le 0.05$. **q** I3P, I3A, and ILA in xenografts of mice fed the No-Trp diet plus vehicle (*n* = 3) or I3P injections (*n* = 3). Each dot represents one mouse. Plots show mean +/− SEM and *P*-value calculated by unpaired *t*-test. *$P \le 0.05$.

## Methods

All experiments adhere rigorously to pertinent ethical regulations and have been both approved and conducted in strict accordance with the guidelines set forth by the Institutional Animal Care and Use Committee at UTSW Medical Center (Protocol Number: 2017–101798) and the Institutional Review Board (IRB) under Protocol STU102010-051.

### Cell culture

All cell lines (human: HUH7, SNU449, HEPG2; mouse: HCC53N) were cultured in Dulbecco's Modified Eagle Medium (DMEM) supplemented with 10% fetal bovine serum (FBS), 1% penicillin-streptomycin. THLE2 cells were cultured in bronchial epithelial growth medium (Lonza) supplemented with 10% FBS. They were moved to DMEM medium for the experiment. Trp-free media was prepared following the manufacturer's instructions (US Biologicals). HCC53N cells were established from mouse liver tumors from Alb-Cre; p53 fl/fl + G12V Nras [35] model. For siRNA transfections, RNAiMAX reagent (Invitrogen) was used in OPTIMEM medium. Cells were seeded at 50,000 cells/well in a 6 well plate and 10,000 cells/well in 24-well plates. After 24 h, siRNAs were transfected at 40 nM following manufacturer's protocol. After 3 days, experiments such as cell viability and qPCR were performed.

### Cell viability

Cells were seeded in 12-well plates, and the next day, the cells were treated with siRNA or drugs at the specified concentration. Cells were fixed with 10% methanol and stained with 0.01% crystal violet solution. The plates were washed 2 times and then distained with 10% acetic acid and the absorbance was measured at 595 nm. Alternatively, cell viability was measured using Apexbio Cell Counting Kit-8 (CCK8). Cells were seeded 2000 cells/well in 96-well plates. The following day, the media was exchanged with 1% FBS Trp-free DMEM/F-12 (NC0181452, Fisher) in accordance with the specified condition. After 72 h, all media was exchanged with 10% CCK8 reagent in media after cells were washed gently with PBS. Plates were returned to the incubator for 2 h before absorbance was measured at 460 nm.

### Mouse studies

Mice were housed at 22 ºC and 30–70% humidity. Mice were fed a chow diet (inotiv: 2916) or special diets control (inotiv: TD.99366), Low-Trp (inotiv: TD.170395) or No-Trp (inotiv: TD.170394). Cages were changed out every other week and health checks were performed by the UTSW veterinary team twice a day. Food and water were topped as needed. MYC-driven liver tumor model mice[48] (FVB background mice expressing TRE-MYC with LAP-tTA) carries MYC as transgene on the Y chromosome, therefore males were used for our experiments. MYC-ON mice were generated by crossing TRE-MYC with LAP-tTA mouse (FVB background). Male breeders with 2 copies of LAP-tTA and single copy of TRE-MYC was crossed with WT FVB females to generate LAP-tTA/

TRE-MYC mice which were used for our experiments. WT FVB males were used as controls. The breeders were maintained on doxycycline water (1 mg/mL). Females in the litters were provided water without doxycycline on the day of birth to activate MYC overexpression. Animals were maintained on a regular chow diet (Envigo). To compare WT and MYC overexpression, FVB WT males were used. WT and MYC overexpression males were 45 days old. For experiments, mice were transferred to either control, Low Trp, or No-Trp diets at the time points indicated in the legends (see diet composition in Table S2). Two weeks after diet change, food consumption was measured using TSE pheno master environmental chambers. Mice were acclimated to these cages for 5 days prior measuring food intake for 4 days. Mice were maintained on a chow diet for 12 weeks and then placed on the control and No-Trp diets. For the Kaplan-Meier curves, the end points were signs of moribund, death, and severe weight loss.

For xenografts, 500,000 HCC53N cells or 3,000,000 HUH7 cells were injected subcutaneously into the flanks of NOD SCID female mice (experiments were also repeated with males and no differences were found). The mice were maintained on a regular chow diet for 3 days. After 5 days, the animals were switched to either the control diet or No-Trp diet. The maximal tumor size permitted by our Institutional Animal Care and Use Committee is ≥2.0 cm$^3$. Mice were euthanized when the tumors reached 2 cm$^3$. For I3P rescue, mice carrying palpable HCC53N xenografts were started on a No-Trp with daily IP injections of I3P at 75 mg/kg or vehicle for 12 days. Mice were sacrificed and weighed, and the tumors were harvested and weighed. For MYC-ON mice, I3P IP injections at 75 mg/kg was administered daily to MYC-ON mice fed a No-Trp diet from days 35 to 48. The mice were sacrificed after 2 weeks of treatment and were weighed, and the livers were harvested and weighed. At the end of each experiment, mice and livers or xenograft tumors were weighed, and blood was obtained. Clotted blood was spun at 15,000 RPM for 10 mins at 4 °C, and the supernatant was collected as serum. Trp and its metabolites were measured in serum and liver. Human samples were ethically sourced from the UT Southwestern Medical Center Biorepository with explicit informed consent for research and publication. Comprehensive details regarding these samples, including self-reported gender, racial and ethnic backgrounds, age at the time of sample collection, and participant counts, are presented in Table S3.

### Trp pathway measurements and in vivo $^{13}$C Trp infusions

LC-MS/MS evaluation of Trp metabolite concentrations was performed by the UTSW Preclinical Pharmacology Core as previously described[13] with the following minor modifications[13]. Mouse and human tissues were homogenized in PBS prior to extraction with 80% final volume methanol. Serum was extracted similarly. Tissue concentrations were normalized to wet tissue weight. $^{13}$C-KA was detected as the 199.962–153.1 transition. AU (arbitrary units) was indicated for metabolites which $^{13}$C-isotope form was not available to generate standard

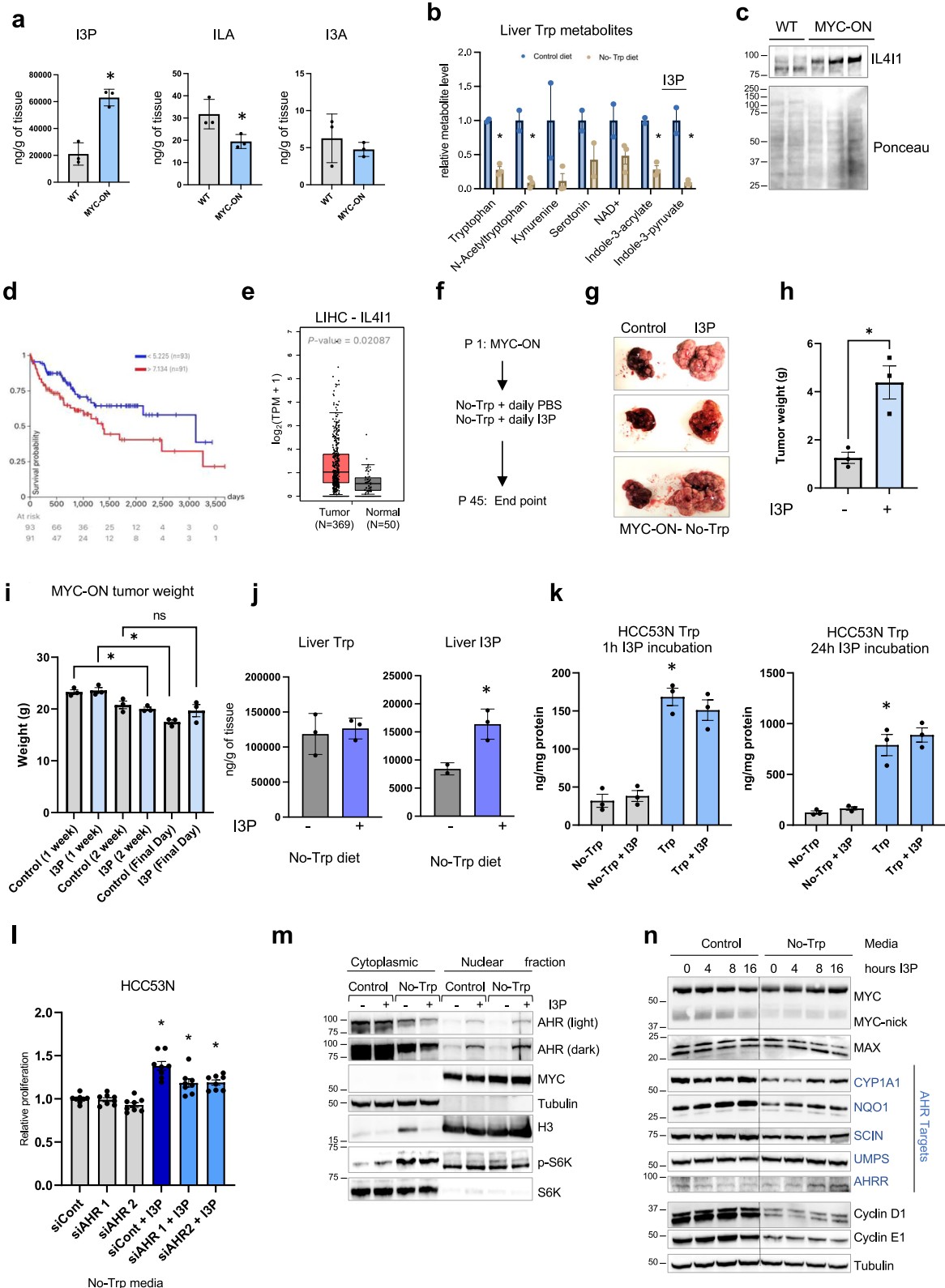

curves and the measurements were calculated based on the $^{12}$C-forms. For $^{13}$C-Trp infusions, 38 days old WT and MYC overexpressing animals were used. Two methods were used to infuse $^{13}$C-Trp. The first method was an intra-peritoneal injection (100 µL of 35 mM $^{13}$C-Trp [Cambridge isotopes]). One h after injection, the animals were euthanized, and their serum and tissues were collected. The second method involved tail vein infusion of $^{13}$C-Trp. Mice were anesthetized and infused with a

35 mM $^{13}$C-Trp; a bolus of 15 mL/min for 10 min and 2 mL/min for 1 h. After the infusion the animals were euthanized, and serum and tissues were collected. Amino acids (other than Trp and its metabolites) were evaluated by LC-MS-MS at the Metabolomics core at Children's Research Institute (UTSW). Tumors samples were processed in 70% methanol and submitted to the core facility to dry the samples in a speedvac and run in LC-MS.

**Fig. 8 | I3P levels are elevated in MYC-ON liver tumors and its supplementation rescues the growth of Trp-starved MYC-ON liver tumors. a** I3P, I3A, ILA in WT ($n = 3$) or MYC-ON ($n = 3$) livers. Each dot represents one mouse. Plots show mean +/− SEM and *P*-value was calculated by unpaired *t*-test. *$P \leq 0.05$. **b** Trp starvation reduces Trp metabolites in the liver. **c** IL4I1 in lysates of WT ($n = 2$) or MYC-ON ($n = 3$) livers. **d** Survival of patients with top 30% (red) and bottom 30% (blue) expression of IL41A. **e** IL41A mRNA in tumors and normal liver from TCGA. (Tumor $n = 369$, Normal $n = 50$) Each dot represents one sample. **f** Experiments with MYC-driven liver tumors fed the No-Trp diet supplemented with I3P. **g** Livers from MYC-ON mice fed the No-Trp diet supplemented with daily injections of either vehicle or I3P. **h** Weigh of livers of MYC-ON mice fed the No-Trp diet supplemented with daily injections of either vehicle ($n = 3$) or (75 μM) of I3P ($n = 3$). Each dot represents one mouse. Plots show mean +/− SEM and *P*-value was calculated by unpaired *t*-test.

*$P \leq 0.05$. **i** Whole-body weight of mice in (**h**). **j** Trp and I3P in MYC-ON livers of animals fed the No-Trp diet supplemented with daily IP injections of either vehicle or I3P (75 μM). **k** Trp in HCC53N cells incubated for 1 h or 24 h with I3P in the presence ($n = 3$) or absence ($n = 3$) of Trp measured by LC-MS/MS. Plot shows mean +/− SEM and *P*-value was calculated by one-way ANOVA. *$P \leq 0.05$. **l** Viability of HCC53N transfected with control siRNA (siCont) ($n = 8$) or siRNA for AHR (siAHR) ($n = 8$) cultured in No- Trp with 20 μM of I3P. Plot shows mean +/− SEM and *P*-value was calculated by one-way ANOVA. *$P \leq 0.05$ (**m**) Western blots of nuclear and cytoplasmic fractionations of HCC53N in the presence or absence of Trp for 1 h supplemented with I3P (25 μM) or DMSO, 3 repeats performed. **n** Western blots of lysates of HCC53N grown in the presence or absence of Trp with I3P (25 μM) or DMSO, 3 repeats performed.

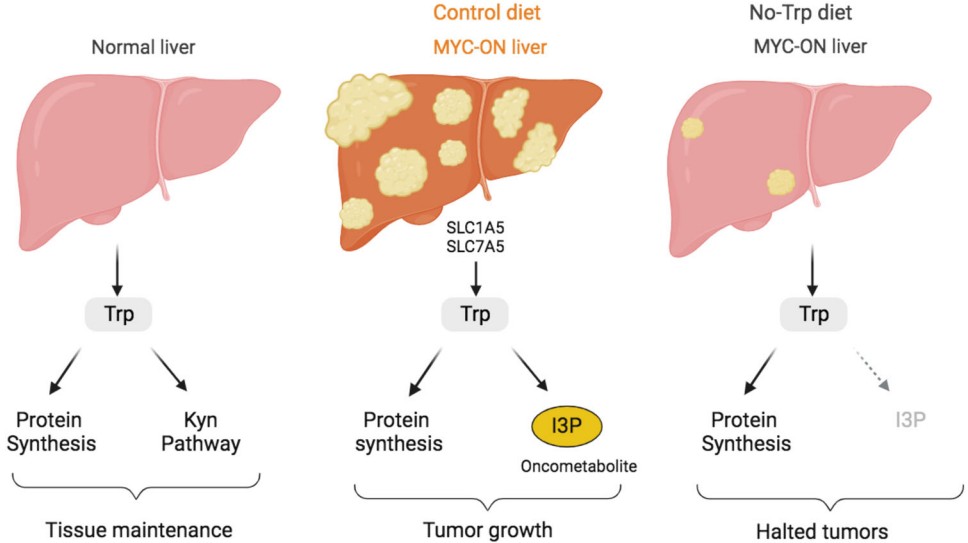

**Fig. 9 | Summary of our findings on Trp dependency and vulnerability in MYC-driven tumors.** MYC upregulation, frequently found in tumors, drives the expression of the Trp transporters SLC1A5 and SLC7A5, which in turn promotes the uptake of Trp. Trp is used to generate the oncometabolite I3P, which activates liver cancer cell growth. This figure was generated with Biorender.com.

## Immunohistochemistry

For IHC livers of WT or MYC-ON mice were fixed and stained as previously described using the anti-MYC antibody ab32072 (Abcam) and anti-KI63 12202 S (Cell Signaling) (see Table S1) at 1:500 diluition[13]. Briefly, tissues were fixed in 10% formalin overnight, embedded in paraffin, and cut 5-μm thick sections. The sections were stained with H&E, and immunohistochemistry performed for MYC and Ki-67. Tissues were deparaffinized with xylene and ethanol. Antigen retrieval was done using the pressure cooker method and blocked with goat serum. Sections were incubated with the primary antibody. Vectastain kit was used and secondary antibody was used at a concentration of 1:500, developed with DAB and mounted with Permount mounting media. Sections were analyzed using the Hamamatsu Nanozoomer and NDP view software.

## TCGA data analysis, RT-qPCR, and RNA-seq analysis

RNA expression from HCC TCGA database was obtained using Xenabrowser. A Log2FC from tumor versus normal paired tissue of 0.568 was used. Survival analysis on HCC patients was performed with oncoLnc database (http://www.oncolnc.org/) with a cut-off of 40% up or downregulation, and Xenabrowser using top and lower quartiles. For RT-qPCR, total RNA was extracted using the Trizol and Qiagen RNA isolation kit. cDNA was produced with Biorad iScript cDNA Synthesis Kit and quantitative PCR with Sybr Green Master Mix (BioRad). *Actin* and *RPS18* were used as housekeeping genes (see Table S1).

RNA-seq was performed by Genewiz/Azenta, and the data processing was done using Trim Galore (https://www.bioinformatics.babraham.ac.uk/projects/trim_galore/) for quality and adapter trimming. The mouse reference genome sequence and gene annotation data, mm10, were downloaded from Illumina iGenomes (https://support.illumina.com/sequencing/sequencing_software/igenome.html). The qualities of RNA sequencing libraries were estimated by mapping the reads onto mouse transcript and ribosomal RNA sequences (Ensembl release 89) using Bowtie (v2.3.4.3). STAR (v2.7.2b) was used to align the reads onto the mouse genome. SAMtools (v1.9) was used to sort the alignments, and HTSeq Python package was used to count reads per gene. DESeq2 R Bioconductor package was used to normalize read counts and identify differentially expressed (DE) genes. KEGG pathway data was downloaded using KEGG API and GO data was downloaded from NCBI FTP. The enrichment of DE genes to pathways and GOs were calculated by Fisher's exact test in R statistical package. The reads were aligned to mouse RefSeq RNA sequences using Burrows-Wheeler Aligner (BWA, v0.7.17) with MEM algorithm and the options, -T 19 -h 200 -Y, and primary alignments with proper read pairs were selected by SAMtools with the options, -f 3 -F 2316, to count the reads aligned to ribosomal RNA sequences defined by the sequence descriptions containing the keyword of ribosomal RNA or rRNA. The reads were aligned to human MYC RNA isoform 1 sequence (NM_002467.6) using BWA with MEM algorithm. The alignments were compared with the alignments to mouse genomic and RNA sequences from NCBI Assembly, GRCm38.p6, using a custom Perl script named

REMOCON[49] (https://github.com/jiwoongbio/REMOCON) and the reads with better alignments to human MYC were counted. To identify transcriptional changes a cut-off of log2FC 1/−1 and adjusted p-value ≤ 0.05 was applied. GO analyses were performed with DAVID software. For transcription factor analyses we implemented same analysis using the transcription factor target data from TRRUST2[50].

## Western blot and proteomics
Proteins were extracted with RIPA lysis buffer with protease and phosphatase inhibitors followed by sonication and centrifugation. Supernatants were run on 4–12% gradient gels and transferred to nitrocellulose membrane to blot with indicated antibodies. Western blots were obtained by chemiluminescence with BioRad ChemiDocImager system. For proteomics, cells/tumors were lysed with RIPA lysis buffer with proteinase inhibitor followed by sonication and centrifugation. Supernatants were run on gels, and proteins extracted for proteomics analysis. MS scans were acquired at 120,000 resolution in the Orbitrap and up to 10 MS/MS spectra were obtained in the ion trap for each full spectrum acquired using higher-energy collisional dissociation (HCD) for ions with charges 2–7. Dynamic exclusion was set for 25 s after an ion was selected for fragmentation. Raw MS data files were analyzed using Proteome Discoverer v2.4 SP1 (Thermo), with peptide identification performed using Sequest HT searching against the mouse protein database from UniProt. Fragment and precursor tolerances of 10 ppm and 0.6 Da were specified, and 3 missed cleavages were allowed. Carbamidomethylation of Cys was set as a fixed modification, with oxidation of Met and either Trp ($^{13}C$ (11)) or Trp converted to Phe set as variable modifications. The false-discovery rate (FDR) cut-off was 1% for all peptides. Peptides identified were assigned to proteins. Proteins identified by at least 1 peptide were used for the analysis. For all proteomics analysis, data was normalized by the sum of all the peptides identified in each sample. After that, for the MYC-liver experiments, a fold change >2 or ≤0.5 and p-value ≤ 0.05 was applied. For the proteins with changes in abundance, Trp percentage for each protein was analyzed and plotted in Fig. 6f. For HCC53N cells experiments, a fold change of >1.3 or ≤0.7 and p-value ≤ 0.05 was applied. Once the cut-off was applied, the gene names of the transcripts was overlapped with the gene names of the proteins with expression changes according to proteomics analysis to identify which protein changes were likely due to changes in mRNA levels or which ones solely changed at protein levels. For Fig. 6b, c, the sum of peak intensities of peptides containing $^{13}C5$-glutamine were normalized with either peak intensities from total peptides or glutamine-containing peptides in each condition and the normalized value was represented in the graph. For Fig. 6g, h, the sum of peak intensities of peptides containing Trp substitution for phenylamine, leucine, isoleucine or tyrosine were normalized with peak intensities from total peptides, the normalized value was represented in the graph.

## Protein synthesis quantification
For puromycylation, cells were starved of FBS overnight, and then FBS-containing media was added for 6 h, followed by puromycin (20 μg/mL) for 2 h. After puromycin incubation, cells were lysed with RIPA buffer for Western blotting using anti-puromycin antibody. Click-IT™ AHA kit incorporation was used following the manufacturer's instructions with some modifications. Cells were starved of FBS and Trp overnight using 1:1 RPMI (without tryptophan Thermo Fisher: 50-190-8106):PBS to reduce methionine levels. The next day, 2% dialyzed FBS with or without Trp together with 100 μM AHA (Thermo Fisher: C10102) was added for 15 min. Click-IT reaction was performed following manufacturer's instructions. Samples were subjected to Western blotting with HRP-streptavidin and imaged with chemiluminescence.

## Statistics and reproducibility
Statistical analyses were performed using two-tailed Student's unpaired t-test, or ANOVA p ≤ 0.05 was considered statistically significant and indicated by asterisks (*) or p value number. The number of samples and biologically independent experiments or technical replicates and associated statistical tests are indicated in the corresponding figure legends. No statistical method was used to pre-determine sample size. Outliers were calculated with GraphPad Prism software and removed if detected. The Investigators were not blinded to allocation during experiments and outcome assessment. Data plotting and statistical analysis were performed using GraphPad Prism. Individual data points are shown for all graphs.

## Reporting summary
Further information on research design is available in the Nature Portfolio Reporting Summary linked to this article.

## Data availability
Source Data are provided with this paper. The RNA-seq comparing control and MYC-on livers from mice fed a control and No-Trp diet data have been deposited in the GEO database under the accession code GSE213377. The data related to heatmaps and plots presented in this study are available within the source data file. The proteomics data obtained from livers of MYC-ON m ice subjected to either a normal or Trp-free diet, as well as from HCC53N tumor xenografts in mice fed the same diets, have been deposited in the MassIVE database (massive.ucsd.edu) under the accession code MSV000089853 (https://massive.ucsd.edu/ProteoSAFe/dataset.jsp?task=ba199ea86ae8423088b62f6740cbf71a). Additionally, the processed proteomics data for both sets of samples are also accessible through MassIVE. For further details, the proteomics data from these samples, as generated in this study, are provided in the Supplementary Information/Source Data file. Source data are provided with this paper.

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

## Acknowledgements

We are grateful to the Sorrell lab members for their valuable feedback and Spencer Barnes for help with TCGA data. The work was financially supported by R35CA22044901to RJD and Cancer Prevention and Research Institute of Texas (CPRIT) (RP220046), American Cancer Society 724003, Welch foundation I-2058-20210327, NCI R01CA245548, NIGMS GM145744-01 and the Circle of friend's award to MCS, NCI R01 CA251928, Simmons Comprehensive Cancer Center Cancer & Obesity Translational Pilot Award, and Mark Foundation 21-003-ELA to HZ, Mary Kay postdoctoral fellowship to PN, and the graduate school training grants CPRIT RP210041 and National Science

Foundation 2022344499 to INB and HHMI Gilliam fellowship program to RG. MCS is John P. Perkins Distinguished Professor in Biomedical Science Virginia Murchison Linthicum Scholar in Medical Research. The authors acknowledge the UT Southwestern institutionally supported Preclinical Pharmacology Core for LC-MS/MS quantitation.

## Author contributions

M.C.S., N.V., R.G., M.L.N., and Y.H.H. planned the experiments and wrote the manuscript. N.V., R.G., M.L.N., A.S., L.P.C., P.A.S.N., I.M., J.K., S.F., I.N.B., L.L., E.M., I.L.S., Y.J., M.B., E.M., performed experiments. I.M. generated HCC53N xenografts. L.K. performs analyses of Trp content in proteins. J.A.K. performed quantitative LC-MS/MS analysis of Trp and its metabolites. J.K. and L.X. performed RNA-seq data analyses. A.L. performed proteomics. J.S., R.J.D., H.Z., N.G., N.S.W., and M.C.S. supervised.

## Competing interests

RJD is a scientific advisor for Agios Pharmaceuticals, Nirogy Therapeutics, Vida Ventures and Droia Ventures, and a founder and advisor for Atavistik Bioscience. The remaining authors declare no other competing interests.
