## [Peer Review File · Nature Communications]

REVIEWER COMMENTS

Reviewer #1 (Remarks to the Author):

In this manuscript, Venkateswaran et al. use RNAseq and proteomics approaches to better understand how hepatocellular carcinoma (HCC) cells utilize tryptophan in an effort to identify metabolic vulnerabilities. They find that while HCC tumors in mice and in humans express higher levels of genes encoding tryptophan transporters, these tumors suppress expression of enzymes related to tryptophan metabolism and show reduced levels of downstream tryptophan metabolites. The authors further show that tryptophan is instead incorporated into proteins in the tumor cells. Finally, they demonstrate in multiple models, both MYC-driven and p53 deletion/mutant NRAS driven, that tryptophan deprivation can slow tumor growth.

The methods used in the paper are thorough and yield a large amount of data. However, the study could be strengthened by corroborating the profiling results with knockout and overexpression experiments to demonstrate the biological relevance of the findings. Additionally, in several instances the authors attempt to draw conclusions that are not well supported by the data presented.

Major points:

1. The majority of the conclusions are drawn from comparative RNAseq data, and upregulated and downregulated genes are described. However, in some of these experiments there are up to four experimental groups (i.e. MYC-on vs WT and control vs 0% tryptophan diet), and it is not clear which groups are being compared to describe the relative gene expression. It is important to clarify this, as both MYC-dependent and MYC-independent effects of tryptophan deprivation are described.
2. A key result observed across HCC models in this paper is that tryptophan is shunted to protein synthesis rather than be metabolized to kynurenine and other downstream intermediates.
 - a. It is suggested that this occurs downstream of MYC via increased expression of tryptophan transporters. However, it is unclear whether this differential partitioning is simply due to increased expression of translation machinery or the suppression of tryptophan metabolism enzymes. This could be tested by overexpression of IDO2, for example, and checking whether this changes tryptophan incorporation into proteins.
 - b. While the authors show that more tryptophan is incorporated into proteins in MYC-on tumors, the total protein content in MYC-on vs WT should also be compared, especially since ribosomal biogenesis is suggested to be altered with tryptophan deprivation.
3. A significant concern is the weight loss phenotype caused by the tryptophan deficient diet.
 - a. It is stated in the text that MYC-on HCC mice survive longer on the 0% tryptophan diet compared to the reduced tryptophan or control diets. However, due to loss of more than 20% of their body weight, the 0% tryptophan mice were required to be euthanized at the same time that the mice on the other diets died. Thus, it would be better to comment on the effects of tryptophan starvation on tumor growth instead of survival.
 - b. A conclusion of the paper is that, under tryptophan deprivation, HCC cells use fatty acids and ketone bodies as fuel. Related to this, it would be useful to characterize the weight loss phenotype in terms of loss of fat vs lean muscle.
 - c. In figure 2n, the authors note that WT mice retain the same liver to body weight ratio between control and 0% tryptophan diets, which is referenced to suggest that the diet is not particularly detrimental. However, since the mice are losing so much weight on the 0% tryptophan diet this ratio cannot be taken as an indicator of health. Either the conclusions from this panel should be better explained in the text or the panel should be removed.
 - d. It is not noted whether mice in the xenograft experiments in figure 3 also exhibit weight loss after 21 days of tryptophan starvation. This should be clarified.
 - e. Because the effects of complete tryptophan deprivation are so detrimental to the animal's health, to increase the potential for clinical application other interventions should be explored that target the vulnerabilities uncovered by tryptophan dependence. For example, since a key use of tryptophan in HCC is protein synthesis, the tumors may be more sensitive to translation inhibition. Alternatively, the tryptophan transporters could be knocked out in the liver as a different strategy to starve the tumors of tryptophan.
 - f. Also related to clinical relevance, the authors note that if advanced tumor-bearing mice were

placed on the 0% tryptophan diet, there was no change in overall survival. However, as this scenario is more analogous to what would happen in the clinic, it would be helpful if tumor growth could be monitored in these animals by ultrasound or if tissues were analyzed at the experiment endpoint to determine whether tryptophan deprivation might have some positive effects.

4. The in vitro results in figure 3, k-n should be repeated in another human HCC cell line in addition to Huh7.

a. The metabolite addback experiments supplementing kynurenine, NA, and NMN in vitro do not add value to the figure because these metabolites have already been shown in this study to be suppressed in HCC. It would be more helpful to measure these metabolites in the cells as they were measured in the MYC-on tumors in figure 1.

b. Can the authors comment on why Huh7 cells were chosen for the xenograft experiments? SNU449 and HEPG2 cells have much higher expression of SLC7A5 at the mRNA level and thus might have stronger responses to tryptophan deprivation.

5. The data in figures 4 and 5 is largely descriptive and stronger biological relevance needs to be established.

a. The gene expression data from both models converges on ribosome biogenesis as a likely cause for reduced tumor growth under tryptophan deprivation. However, in figure S5 the authors note that nucleolar stress, an indicator of decreased ribosomal biogenesis, is not observed. Without other evidence, this calls into question whether the gene expression data actually aligns with reduced ribosomal biogenesis. Other evidence for reduced ribosome biogenesis should be presented.

b. In figure 4, it should be clarified whether the bar graphs of gene expression data are representing the RNAseq data or qPCR validation of the RNAseq results.

c. Since tryptophan is so detrimental to the organism, it would be useful to know if blocking expression of any of the genes shown in figures 4 and 5 is sufficient to limit tumor growth. This would point to a potential alternative clinical approach and strengthen the mechanistic basis for tryptophan deprivation reducing tumor growth.

Minor points:

1. When describing the setup of the diet experiments, the age of the mice when the diet started should be noted. As written and laid out in the diagrams, it is not clear whether the mice started the tryptophan diets right after weaning or at a different time.

2. Throughout the figures, significance stars are frequently the same size and color as the other points on the graph and are therefore difficult to distinguish. Consider changing the size and/or color to make these clearer.

3. There is reference to a "figure S2H" but this panel does not exist in the figure.

4. On page 9 in paragraph 1, the authors write, "Therefore, we surmised that Trp deprivation could be an efficient approach to limit the availability of both Trp and all its catabolites." However, the authors have already noted that tryptophan catabolites are already suppressed consistently across HCC models, and thus the tumors do not need to be deprived of catabolites downstream of tryptophan. This sentence can be deleted.

5. On page 11 in paragraph 2, the authors state that the HCC53N model was used to study MYC-independent effects of tryptophan starvation.

a. Is p53 deletion and mutant NRAS a common mutation in human HCC? This should be commented upon.

b. While these tumors are not exclusively MYC driven, in figure 4j it appears these cells might have elevated MYC levels. MYC should be blotted for in these cells alongside samples from WT mice to clarify this.

c. In the same paragraph, the authors write "This suggests that Trp starvation causes a reduction in ribosome biogenesis in tumors initiated by different oncogenic lesions and that pathways downstream of MYC (or other initiating mutations) are likely the main cause of growth inhibition by this diet." This sentence is confusing because it was suggested previously that HCC53N cells are not dependent on MYC, and further no signaling data is presented as far as what pathways downstream of NRAS might be altered. Thus, this statement must be clarified or deleted.

6. In figure 4i, the labels for the western blot lanes do not line up with the samples.

7. The text size on the graph in figure 5b on the right is far too small. Please make larger.

8. On page 12 in the last sentence of paragraph 2, the authors note, "In contrast, MLXIPL levels were not significantly upregulated." Then, in the start of paragraph 4, they write, "MLXIPL, which was inferred to be activated in both tumor models upon Trp starvation..." Please clarify the

discrepancy between these two statements.

9. In figure 6, it is shown that with tryptophan deprivation both MYC-on and HCC53N tumors have reduced levels of metabolites related to fatty acid oxidation and reduced ketones. In the discussion of the figures, the authors use this data to claim that these tumors are using fatty acids and ketones as fuel. However, the metabolite data alone is not sufficient to make this claim; tracing of a labeled substrate or other approach needs to be used.

Reviewer #2 (Remarks to the Author):

Venkateswaran et al. present a study examining the metabolism of the amino acid tryptophan in models of MYC-driven liver cancer (HCC). Motivated by RNAseq data showing changes in tryptophan transporter and metabolism enzyme expression in MYC driven HCC, the authors explored the fate of tryptophan in HCC and tumor dependencies on tryptophan. Tryptophan is the least encoded for proteogenic amino acid but is also a precursor to the de novo synthesis of niacin (needed for nicotinamide dinucleotide synthesis) and intermediates in tryptophan metabolism (such as kynurenine) have well-documented roles in cancer cell signaling. In HCC, the authors observed that increased tryptophan intake was not correlated with increased tryptophan metabolism but rather appeared to be fueling protein synthesis. Placing tumor bearing mice on a tryptophan free diet slowed tumor growth significantly, albeit in the context of significant long-term toxicity from the amino acid deprivation itself (animals were restricted to no trp diet for 33 days due to weight loss- this was already significant at 21 days). The authors showed that in human HCC patients, there was no increase in kynurenine either, a finding in contrast to that of CRC patients. The effects of tryptophan restrictions on tumor growth could be attributed to a general arrest of the cell cycle as well as ribosome biogenesis and translation. Finally, tumors cultured in TRP free conditions showed an increase dependence on fatty acid oxidation and in nod-SCID animals harboring xenograft HCC tumors, the effects of tryptophan restriction were rescued by high fat diet (as was to some degree animal weight).

The strengths of this study are the consistent results across models (MYC-on in vivo tumors, xenografts and cell culture studies) and the quality of the data from the metabolomics and transcriptomics characterization. The authors also have clear data showing that tryptophan metabolism per se is not a driving factor here, but rather it appears that the use of this amino acid for protein synthesis is the most important mechanism at play. That said, tryptophan is an essential amino acid, and complete restriction was not well-tolerated by animals. It is unclear how unique tryptophan dependency is compared to other essential amino acids in this tumor type (phenylalanine or leucine for example). This reviewer feels the studies are well-done and the results interesting, but that the broader significance of this work requires more study. I think the authors need to address the question of whether this result is solely due to restricting an essential amino acid and tryptophan is convenient, or whether tryptophan is in some way unique?

Comments/ Suggestions

- 1) The primary mechanistic hypothesis of the tryptophan effect on tumor growth is direct restriction of protein synthesis. This finding would be significantly improved by direct biochemical evidence of amino acid stress, e.g. tRNA charging, GCN2 activation etc. Data presented on codon substitutions does report on significant unresolved AA stress, but does not report on the upstream known signaling pathways that in mammals induce stress pathways when AA are lacking and that exist in part to mitigate protein AA misincorporation. Direct characterization of protein synthesis rates (using S35 for example) would also help quantify the effect of tryptophan restriction in vivo.
- 2) Do similar effects happen with other essential and nonessential amino acids? In multiple places in the manuscript, there is an opportunity to probe the use of other amino acids and explore how unique tryptophan dependency is. For example, in cell lines, does phenylalanine or isoleucine restriction phenocopy tryptophan? In mouse and human liver samples, are other essential amino acids also altered?
- 3) Similarly, the authors argue that MYC is important for this effect. But given that tryptophan is an essential amino acid, are MYC neg cells really resistant to tryptophan starvation? THLE cells are a good representation of non-cancerous cells, but what about additional cancer cell lines? Have they enhanced tryptophan uptake as well, but by MYC independent mechanisms?

4) I am concerned about the significant and critical weight loss that the tryptophan deficient diet induced in mice. Can the authors control for the systemic effects of malnutrition on tumor growth? Phrased differently, could tumors be growing poorly for cell extrinsic reasons related to effects on the immune system or other aspect physiology due to chronic and symptomatic tryptophan deficiency?

5) In the HFD rescue experiment, it was noted that HFD also partially rescued weight loss. Might healthier animals have healthier tumors? The authors should provide biochemical evidence to characterize protein synthesis or other markers of tryptophan deficiency stress in these HFD-trp tumors. If HFD can alleviate these how so? And if tumors are as protein stressed as before, is their enhanced macroscopic growth due to hyperplasia or hypertrophy engendered by HFD?

Reviewer #3 (Remarks to the Author):

In this manuscript, the authors describe an increased dependence on tryptophan uptake in liver cancer and the consequent therapeutic potential of dietary tryptophan restriction.

The first half of the paper contains some striking data on tryptophan metabolism in HCC models and the authors make a good case that elevated tryptophan import is used to support the increased biosynthetic (particularly translational) drive promoted by oncogenic MYC. Additional cell HCC models (including one driven by Ras/p53-loss) show similar dependence upon dietary tryptophan and in the MYC-driven and xenograft experiments, there is a strong reduction in tumour growth upon dietary tryptophan restriction. The authors also present some patient data showing similar neutral amino acid (NAA) transporter expression patterns in HCC. As might be expected for removal of an essential amino acid, there are strong reductions in gene signatures for cell cycle, DNA replication and ribosome biogenesis. This section is interesting and informative and the novelty is only slightly tempered by the authors' previous publication describing elevated tryptophan uptake driven by MYC in colon cancer cells. For this part of the manuscript, it would be interesting to know if deprivation of other essential amino acids elicits similar effects, or whether there is something unique to tryptophan. The use of inhibitors of the NAA transporters would also add some therapeutic value.

The second half of the manuscript deals with transcriptional and proteomic changes that implicate fatty acid (FA)/lipid catabolism in support of HCCs undergoing tryptophan deprivation. I found this link very tenuous with minimal metabolic evidence for FA oxidation and the only perturbation experiment a rescue of HCC growth by high-fat diet. Several lipid biosynthesis genes are upregulated and SREBF1 gives a strong signature, which is hard to reconcile with a switch to lipid catabolism and the simplest (in my view) explanation for the appearance of ketone bodies upon tryptophan deprivation is the catabolism of the ketogenic amino acids leucine and lysine due to a lack of global translation. Furthermore, I don't think that the reduction in the absolute levels of carnitines and acylcarnitines presented here are necessarily diagnostic of reduced FA oxidation. The inclusion of experiments utilising etomoxir and providing direct evidence for induction of FA oxidation upon tryptophan withdrawal would have been necessary first steps in this direction. There is also no mechanistic insight (that I can see) into how tryptophan deprivation might lead to reprogramming of lipid metabolism. If the authors wish to make the link with lipid catabolism, this would have to be greatly consolidated in my view.

Minor points:

Please spell out kynurenine (or define as an abbreviation) throughout.

Figs 4,5 - The small numbers in the pathway analyses must be explained in the legend – number of genes identified in the pathway?

Combined analysis is not explained in methods, promoter analysis is not explained in methods.

In the promoter analyses (Z-scores), why are there zero values and why are they included?

Fig 4d-g,m should be in supplementary data (if included at all).

Source of cell lines and of mice (especially transgenics) is not given. Why were littermates lacking the TRE-Myc not used as controls in place of wt FVB? Indeed, in Fig 2n, the text says WT mice, but the figure says 'Myc Off'. Please check this.

In methods, the immunohistochemistry section lacks several details that would be needed to allow reproduction of the experiments. Furthermore, no antibody information is given.

Please correct 'acetylcarnitines' to acylcarnitines in the text.

nM/mg should presumably be nmol/mg on several axes. Furthermore AU appears on some axes – please clarify what this means.

'When Trp was re-introduced to the diet of these mice, the tumors rapidly grew (Fig S2G-H)' - this is not shown (only viability) – please rephrase. Also, include H as the panel title.

Please improve the resolution of Fig S3A and increase font for p-value.

RESPONSE TO REVIEWERS' COMMENTS

We are pleased to submit our revised manuscript titled "Tryptophan Fuels MYC-Dependent Liver Tumorigenesis through Indole 3-Pyruvate Synthesis." This revised version incorporates significant improvements based on the critiques and suggestions from reviewers 1, 2, and 3.

Our revised manuscript highlights the following key findings:

- MYC-driven liver tumors rely on increased uptake of tryptophan (Trp) for their growth.
- Deprivation of Trp through a Trp-free diet effectively prevents tumor growth.
- Protein synthesis remains unaffected in liver cancer cells despite Trp starvation, revealing the compensatory upregulation of the Trp transporter SLC1A5 in Trp-starved liver tumors.
- Indole 3-pyruvate (I3P), a metabolite derived from Trp, is the omcometabolite mediating growth-promoting functions of Trp in vivo. Supplementation of I3P successfully restores the growth of Trp-starved liver tumors.

Significant developments in experiments and figures include:

- **Revised Figure 2:** A comprehensive analysis of the side effects associated with Trp deprivation, in response to the suggestions from Rev# 1.
- **New Figure 4:** Combined analysis of all RNA-seq datasets as requested by Rev#1.
- **New Figure 6:** Protein synthesis measured in vitro and in vivo was unaffected under Trp-deprived conditions; addressing the concerns raised by Rev# 1 and 2.
- **New Figure 7 and 8:** Characterize the identification of the lesser-known Trp metabolite, I3P, as the key mediator of growth-promoting functions downstream of Trp in liver cancer cells.
- **The section on the rescue of Trp-starved tumors by a high-fat diet, initially presented in our previous submission, has been removed.** This decision is in response to concerns raised by Reviewer #3 and will be addressed separately to elucidate the molecular mechanism underlying the rescue effect of a high-fat diet.

Reviewer #1 comments

Major comments:

Reviewer #1: 1. The majority of the conclusions are drawn from comparative RNAseq data, and upregulated and downregulated genes are described. However, in some of these experiments there are up to four experimental groups (i.e. MYC-on vs WT and control vs 0% tryptophan diet), and it is not clear which groups are being compared to describe the relative gene expression. It is important to clarify this, as both MYC-dependent and MYC-independent effects of tryptophan deprivation are described.

Our response: We thank the review for this suggestion. Now, we present one single figure (**Figure 4**) containing RNA-seq data analyses comparing MYC activated and repressed genes with genes affected by Trp starvation. We found that Trp starvation reverts transcriptional profiles driven by MYC to better resemble transcriptional profile of WT livers.

Reviewer #1: 2. A key result observed across HCC models in this paper is that tryptophan is shunted to protein synthesis rather than be metabolized to kynurenine and other downstream intermediates. a. It is suggested that this occurs downstream of MYC via increased expression of tryptophan transporters. However, it is unclear whether this differential partitioning is simply due to increased expression of translation

machinery or the suppression of tryptophan metabolism enzymes. This could be tested by overexpression of IDO2, for example, and checking whether this changes tryptophan incorporation into proteins.

b. While the authors show that more tryptophan is incorporated into proteins in MYC-on tumors, the total protein content in MYC-on vs WT should also be compared, especially since ribosomal biogenesis is suggested to be altered with tryptophan deprivation.

Our response: Once again, this was an insightful suggestion by the reviewer and generated an entire new Figure with extensive experiments designed to directly test the effects of Trp starvation in translation capacity of liver cancer cells both *in vivo* and *in vitro* (new **Figures 6 A-J and Supplementary Figure S6 A-K**). Surprisingly, we found that Trp starvation does not reduce protein synthesis in Trp starved liver cancer cells and tissues. Importantly, we found that upon Trp starvation, the expression of the Trp transporter SLC1A5 was elevated, thus suggesting that Trp starvation does not reduce translation, likely due to an increase in Trp uptake. Moreover, Trp is the least abundant amino acid in the proteome. These findings led us to investigate the role of other metabolites downstream of Trp in driving growth.

Reviewer #1: 3. A significant concern is the weight loss phenotype caused by the tryptophan deficient diet. a. It is stated in the text that MYC-on HCC mice survive longer on the 0% tryptophan diet compared to the reduced tryptophan or control diets. However, due to loss of more than 20% of their body weight, the 0% tryptophan mice were required to be euthanized at the same time that the mice on the other diets died. Thus, it would be better to comment on the effects of tryptophan starvation on tumor growth instead of survival.

Our response: We have now clarified that the effects on tumor growth (**Figure 3G-K**). These mice were fed a complete or Tr-depleted diet and all sacrificed at day 50. We found a reduction in tumor burden with Trp starvation.

Reviewer #1: b. A conclusion of the paper is that, under tryptophan deprivation, HCC cells use fatty acids and ketone bodies as fuel. Related to this, it would be useful to characterize the weight loss phenotype in terms of loss of fat vs lean muscle.

Our response: We have removed the fatty acid component, because we do not yet understand the molecular mechanisms by which Trp starvation is rescued by high fat diet. We will explore this finding in a separate study. The new **Figure 2** shows a characterization of the side effects caused by Trp starvation in animals' weight, body composition, respirations rate and heat production. Our experiments suggest that Trp starvation caused a preferential reduction in body fat and preserved lean mass, (**Figure 2J-S**).

Reviewer #1: c. In figure 2n, the authors note that WT mice retain the same liver to body weight ratio between control and 0% tryptophan diets, which is referenced to suggest that the diet is not particularly detrimental. However, since the mice are losing so much weight on the 0% tryptophan diet this ratio cannot be taken as an indicator of health. Either the conclusions from this panel should be better explained in the text or the panel should be removed.

Our response: We have rephrased this paragraph to describe the observed phenotype in terms of tumor growth and to avoid any over-interpretations about this ration as a reflection of overall health. We now say "The liver weight of mice with MYC-driven tumors reached 40% of the body weight. However, the liver weight of mice fed the No-Trp diet was around 10% of their body weight (Figure. 3c, d). Moreover, the No-Trp diet did not affect the liver-body ratio in WT mice (Fig. 3d), indicating that Trp reduction has a more severe effects

on cancer cells in the liver. We confirmed that the levels of Trp and Kyn (Figure 3e-f) were lower with Trp starvation in the MYC-driven liver tumors”

Reviewer #1 d. It is not noted whether mice in the xenograft experiments in figure 3 also exhibit weight loss after 21 days of tryptophan starvation. This should be clarified.

Our response: The weight loss of NODSCID mice was added to **Figure S4i**. We did normalize tumor/body weight because unfortunately we did not record the weight of the same mice where the tumors were grown.

Reviewer #1: e. Because the effects of complete tryptophan deprivation are so detrimental to the animal’s health, to increase the potential for clinical application other interventions should be explored that target the vulnerabilities uncovered by tryptophan dependence. For example, since a key use of tryptophan in HCC is protein synthesis, the tumors may be more sensitive to translation inhibition. Alternatively, the tryptophan transporters could be knocked out in the liver as a different strategy to starve the tumors of tryptophan.

Our response: Given that Trp starvation did not affect translation, contrary to our expectation, we do not predict that inhibition of protein synthesis is the mechanisms by which Trp starvation blocks tumor growth. We now show that I3P is the oncometabolite generated from Trp that can rescue the growth of Trp-starved tumors (**Figures 7-8**). Thus, bringing specificity to the molecular mechanism by which Trp drives tumor growth.

Reviewer #1: f. Also related to clinical relevance, the authors note that if advanced tumor-bearing mice were placed on the 0% tryptophan diet, there was no change in overall survival. However, as this scenario is more analogous to what would happen in the clinic, it would be helpful if tumor growth could be monitored in these animals by ultrasound or if tissues were analyzed at the experiment endpoint to determine whether tryptophan deprivation might have some positive effects.

Our response: We have clarified that placing animals with advanced disease on a Trp-free diet has a more modest effect on tumor growth (**Figure 3L-m**). We were unable to monitor tumor growth in individual mice, because of malfunctioning equipment in our mouse facility.

Reviewer #1: 4. The in vitro results in figure 3, k-n should be repeated in another human HCC cell line in addition to Huh7.

Our response: The reason why we chose Huh7 cells is that their tumors form more rapidly. We have attempted the other lines, and they do not grow in mice in the absence of Matrigel. The extended time for growth would require that the mice stay on a Trp-free diet for a very long time, which would cause side effects. Moreover, we do not include Matrigel in the injection, because it may become a source of amino acids and damper the effects of Trp starvation in tumors. We observe reduction of tumor growth in xenografts on Huh7 and HCC53N cells as well and in the MYC driven mouse model of liver cancer. These are from diverse background, thus showing that the effects of Trp starvation and I3P rescue are likely widespread.

Reviewer #1 a. The metabolite addback experiments supplementing kynurenine, NA, and NMN in vitro do not add value to the figure because these metabolites have already been shown in this study to be suppressed in HCC. It would be more helpful to measure these metabolites in the cells as they were measured in the MYC-on tumors in figure 1.

Our response: Following this recommendation, we found that the Trp metabolite I3P (which has been little studied) was elevated in tumors, and we found that I3P is extremely potent at rescuing the growth of Trp-starved liver cancer (**Figures 7-8**).

Reviewer #1 b. Can the authors comment on why Huh7 cells were chosen for the xenograft experiments? SNU449 and HEPG2 cells have much higher expression of SLC7A5 at the mRNA level and thus might have stronger responses to tryptophan deprivation.

Our response: See our answer above. We tried all 3 cells lines but found that SNU and HEPG2 grow very slowly (taking many months) and require Matrigel for growth, which makes these cells not ideal for our experiments. Therefore, we chose Huh7 cells.

Reviewer #1: 5. The data in figures 4 and 5 is largely descriptive and stronger biological relevance needs to be established.

Our response: We agree with the reviewer. We present RNA-seq data analyses in one single Figure (**Figure 4**) that demonstrate that Trp starvation causes transcriptional changes that make MYC-ON liver expression profiles resemble WT liver transcriptional patterns. New Figures 7 and 8 are fully dedicated to defining the molecular mechanism by which Trp promotes tumor growth.

Reviewer #1: a. The gene expression data from both models converges on ribosome biogenesis as a likely cause for reduced tumor growth under tryptophan deprivation. However, in figure S5 the authors note that nucleolar stress, an indicator of decreased ribosomal biogenesis, is not observed. Without other evidence, this calls into question whether the gene expression data actually aligns with reduced ribosomal biogenesis. Other evidence for reduced ribosome biogenesis should be presented.

Our response: No differences in translation were observed.

Reviewer #1 b. In figure 4, it should be clarified whether the bar graphs of gene expression data are representing the RNAseq data or qPCR validation of the RNAseq results.

Our response: The legends now clearly state whether the bar graphs plotted from RNA-seq or from qPCR.

Reviewer #1 c. Since tryptophan is so detrimental to the organism, it would be useful to know if blocking expression of any of the genes shown in figures 4 and 5 is sufficient to limit tumor growth. This would point to a potential alternative clinical approach and strengthen the mechanistic basis for tryptophan deprivation reducing tumor growth.

Our response: Blocking the enzyme that generates I3P may provide similar effects to Trp starvation. Our future studies will focus on determining if IL4I1 (documented to generate I3P from Trp) is the sole enzyme generating I3P and developing genetic and pharmacologic tools to inhibit I3P production.

Minor points:

Reviewer #1 1. When describing the setup of the diet experiments, the age of the mice when the diet started should be noted. As written and laid out in the diagrams, it is not clear whether the mice started the tryptophan diets right after weaning or at a different time.

Our response: We have now clarified in the scheme for each experiment when the dietary changes were introduced. See **Figure 3a, l, m, Figure 8f** for examples.

Reviewer #1 2. Throughout the figures, significance stars are frequently the same size and color as the other points on the graph and are therefore difficult to distinguish. Consider changing the size and/or color to make these clearer.

Our response: We have changed the size of the significance stars.

Reviewer #1 3. There is reference to a “figure S2H” but this panel does not exist in the figure.

Our response: We have corrected this mistake

Reviewer #1: 4. On page 9 in paragraph 1, the authors write, “Therefore, we surmised that Trp deprivation could be an efficient approach to limit the availability of both Trp and all its catabolites.” However, the authors have already noted that tryptophan catabolites are already suppressed consistently across HCC models, and thus the tumors do not need to be deprived of catabolites downstream of tryptophan. This sentence can be deleted.

Our response: With the finding that I3P is elevated in liver tumors, it is likely that I3P starvation is the cause of tumor reduction.

Reviewer #1: 5. On page 11 in paragraph 2, the authors state that the HCC53N model was used to study MYC-independent effects of tryptophan starvation.a. Is p53 deletion and mutant NRAS a common mutation in human HCC? This should be commented upon.

Our response: Our text now reads: “To determine whether tumors arising from different genetic alterations were also dependent on higher Trp levels and were sensitive to Trp starvation, we generated a primary mouse liver cancer cell line HCC53N³⁵ from tumors driven by the combination of p53 KO and the overexpression of mutant N-Ras both genetic alterations found to occur in HCC^{36,37}.”

Reviewer #1 b. While these tumors are not exclusively MYC driven, in figure 4j it appears these cells might have elevated MYC levels. MYC should be blotted for in these cells alongside samples from WT mice to clarify this.

Our response: This is shown on a new western blot in **Figure 5J**. HCC53N and HUH7 express higher levels of MYC than WT livers, which was expected given that more than 90% of all human tumors have an elevation in MYC.

Reviewer #1 c. In the same paragraph, the authors write “This suggests that Trp starvation causes a reduction in ribosome biogenesis in tumors initiated by different oncogenic lesions and that pathways downstream of MYC (or other initiating mutations) are likely the main cause of growth inhibition by this diet.” This sentence is confusing because it was suggested previously that HCC53N cells are not dependent on MYC, and further no signaling data is presented as far as what pathways downstream of NRAS might be altered. Thus, this statement must be clarified or deleted.

Our response: This section has been extensively edited

Reviewer #1: 6. In figure 4i, the labels for the western blot lanes do not line up with the samples.

Our response: This has been corrected.

Reviewer #1: 7. The text size on the graph in figure 5b on the right is far too small. Please make larger.

Our response: We have ensured that all labels in this edited version are visible.

Reviewer #1 8. On page 12 in the last sentence of paragraph 2, the authors note, "In contrast, MLXIPL levels were not significantly upregulated." Then, in the start of paragraph 4, they write, "MLXIPL, which was inferred to be activated in both tumor models upon Trp starvation..." Please clarify the discrepancy between these two statements.

Our response: This section was removed from the new version of the manuscript.

Reviewer #1 9. In figure 6, it is shown that with tryptophan deprivation both MYC-on and HCC53N tumors have reduced levels of metabolites related to fatty acid oxidation and reduced ketones. In the discussion of the figures, the authors use this data to claim that these tumors are using fatty acids and ketones as fuel. However, the metabolite data alone is not sufficient to make this claim; tracing of a labeled substrate or other approach needs to be used.

Our response: This section was been removed from the new version of the manuscript.

Reviewer #2 comments

Reviewer #2: "This reviewer feels the studies are well-done and the results interesting, but that the broader significance of this work requires more study. I think the authors need to address the question of whether this result is solely due to restricting an essential amino acid and tryptophan is convenient, or whether tryptophan is in some way unique?"

Our response: Our revised manuscript demonstrates that the Trp metabolite I3P is the driver of tumors in the liver of MYC-expressing livers. Given that I3P can only be generated from Trp it is not likely that other amino acids drive tumor growth via the same mechanism as Trp. Nevertheless, we will compare starvation of each essential amino acid in the growth of MYC-driven liver tumors as a future study.

Comments/ Suggestions

Reviewer #2: 1) The primary mechanistic hypothesis of the tryptophan effect on tumor growth is direct restriction of protein synthesis. This finding would be significantly improved by direct biochemical evidence of amino acid stress, e.g. tRNA charging, GCN2 activation etc. Data presented on codon substitutions does report on significant unresolved AA stress, but does not report on the upstream known signaling pathways that in mammals induce stress pathways when AA are lacking and that exist in part to mitigate protein AA misincorporation. Direct characterization of protein synthesis rates (using S35 for example) would also help quantify the effect of tryptophan restriction in vivo.

Our response: We performed extensive experiments designed to directly test the effects of Trp starvation of the translation capacity of liver cancer cells both in vivo and in vitro (new **Figures 6 A-J** and Supplementary **Figure S6 A-K**). Surprisingly, we found that Trp starvation does not reduce protein synthesis in Trp-starved liver cancer cells and tissues. Importantly, we found that upon Trp starvation, the expression of the Trp transporter SLC1A5 was elevated, thus suggesting that Trp starvation does not reduce translation, in part, due to an increase in Trp uptake. Moreover, Trp is the least abundant amino acid in the proteome. These findings led us to investigate the role of other metabolites downstream of Trp in driving growth.

Reviewer #2: 2) Do similar effects happen with other essential and nonessential amino acids? In multiple places in the manuscript, there is an opportunity to probe the use of other amino acids and explore how unique tryptophan dependency is. For example, in cell lines, does phenylalanine or isoleucine restriction phenocopy tryptophan? In mouse and human liver samples, are other essential amino acids also altered?

Our response: Our revised manuscript demonstrates that the Trp metabolite I3P is the driver of tumors in the liver of MYC-expressing livers (**Figures 7-8**). Given that the indole ring in I3P can only be generated from Trp it not likely that other amino acids drive tumor growth via the same mechanism as Trp. Nevertheless, we will compare starvation of each essential amino acid in the growth of MYC-driven liver tumors in future studies.

Reviewer #2: 3) Similarly, the authors argue that MYC is important for this effect. But given that tryptophan is an essential amino acid, are MYC neg cells really resistant to tryptophan starvation? THLE cells are a good representation of non-cancerous cells, but what about additional cancer cell lines? Have they enhanced tryptophan uptake as well, but by MYC independent mechanisms?

Our response: In our experience, cells lines have different sensitivity to Trp starvation; normal cells withstand longer periods than transformed cells. We agree with the reviewer that long-term Trp starvation is likely to affect normal cells. While we have performed experiments in other cancer cell types and are extending our studies to in vivo GEMM and xenograft models, we chose to focus on liver cancer here because we have indication that different tumor types rely on different Trp metabolites to grow. We can conclusively say that liver tumors rely on I3P, but we are in the process on investigating the dependencies of other tumor types.

Reviewer #2: 4) I am concerned about the significant and critical weight loss that the tryptophan deficient diet induced in mice. Can the authors control for the systemic effects of malnutrition on tumor growth? Phrased differently, could tumors be growing poorly for cell extrinsic reasons related to effects on the immune system or other aspect physiology due to chronic and symptomatic tryptophan deficiency?

Our response: it is possible that Trp starvation has physiological non-cell autonomous effects that contribute to reduced tumor growth. Future experiments will address the involvement of the immune system in these effects. However, it is worth mentioning that Trp starvation reduces the growth of xenotransplanted tumors in immunocompromised mice. Moreover, we now know that I3P, which does not affect body weight, is the oncometabolite downstream of Trp, suggesting a more specific effect.

Reviewer #2: 5) In the HFD rescue experiment, it was noted that HFD also partially rescued weight loss. Might healthier animals have healthier tumors? The authors should provide biochemical evidence to characterize protein synthesis or other markers of tryptophan deficiency stress in these HFD-trp tumors. If HFD can alleviate these how so? And if tumors are as protein stressed as before, is their enhanced

macroscopic growth due to hyperplasia or hypertrophy engendered by HFD?

Our response: Given the limited knowledge we have about the molecular mechanism causing rescue of Trp starvation by HFD, we removed this section and will develop it as an independent study. Our revised manuscript now shows that reduction in tumor growth caused by Trp starvation can be rescued by I3PA and that I3PA does not prevent weight loss caused by Trp starvation, decoupling weight loss and tumor growth.

Reviewer #3 (Remarks to the Author):

Reviewer #3: In this manuscript, the authors describe an increased dependence on tryptophan uptake in liver cancer and the consequent therapeutic potential of dietary tryptophan restriction.

The first half of the paper contains some striking data on tryptophan metabolism in HCC models and the authors make a good case that elevated tryptophan import is used to support the increased biosynthetic (particularly translational) drive promoted by oncogenic MYC. Additional cell HCC models (including one driven by Ras/p53-loss) show similar dependence upon dietary tryptophan and in the MYC-driven and xenograft experiments, there is a strong reduction in tumour growth upon dietary tryptophan restriction. The authors also present some patient data showing similar neutral amino acid (NAA) transporter expression patterns in HCC. As might be expected for removal of an essential amino acid, there are strong reductions in gene signatures for cell cycle, DNA replication and ribosome biogenesis. This section is interesting and informative and the novelty is only slightly tempered by the authors' previous publication describing elevated tryptophan uptake driven by MYC in colon cancer cells. For this part of the manuscript, it would be interesting to know if deprivation of other essential amino acids elicits similar effects, or whether there is something unique to tryptophan. The use of inhibitors of the NAA transporters would also add some therapeutic value.

Our response: The reviewer is correct. Our body of work shows that colon cancer and liver cancer tumors contain different ratios of Trp metabolites. Colon cancer has an increase in Kyn while liver cancer shows a decrease. We have concluded that this is related to the intrinsic role of the liver in metabolizing Trp all the way down to NAD⁺. Given that the mechanism by which Trp induces liver tumor growth is likely via the generation of I3P it is unlikely that other amino acids will cause the same effect. Nevertheless, we will investigate the effects of starvation of every single amino acid in the growth of Myc-driven liver tumors in future studies.

Reviewer #3: The second half of the manuscript deals with transcriptional and proteomic changes that implicate fatty acid (FA)/lipid catabolism in support of HCCs undergoing tryptophan deprivation. I found this link very tenuous with minimal metabolic evidence for FA oxidation and the only perturbation experiment a rescue of HCC growth by high-fat diet. Several lipid biosynthesis genes are upregulated and SREBF1 gives a strong signature, which is hard to reconcile with a switch to lipid catabolism and the simplest (in my view) explanation for the appearance of ketone bodies upon tryptophan deprivation is the catabolism of the ketogenic amino acids leucine and lysine due to a lack of global translation. Furthermore, I don't think that the reduction in the absolute levels of carnitines and acylcarnitines presented here are necessarily diagnostic of reduced FA oxidation. The inclusion of experiments utilising etomoxir and providing direct evidence for induction of FA oxidation upon tryptophan withdrawal would have been necessary first steps in this direction. There is also no mechanistic insight (that

I can see) into how tryptophan deprivation might lead to reprogramming of lipid metabolism. If the authors wish to make the link with lipid catabolism, this would have to be greatly consolidated in my view.

Our response: In response to the reviewers concern, we removed this section of the manuscript and focused the paper on the molecules and pathways downstream of Trp-driven tumor growth. The cross talk between lipid metabolism and Trp starvation requires further investigation and will be developed into a separate study.

Minor points:

Reviewer #3: Please spell out kynurenine (or define as an abbreviation) throughout.

Our response: We have spelled out Kynurenine in the abstract

Reviewer #3: Figs 4,5 - The small numbers in the pathway analyses must be explained in the legend – number of genes identified in the pathway?

Our response: We have corrected these issues.

Reviewer #3: Combined analysis is not explained in methods, promoter analysis is not explained in methods. In the promoter analyses (Z-scores), why are there zero values and why are they included?

Our response: We have now extended the description of the analyses including an explanation below: “To identify transcriptional changes a cut off of \log_2FC 1/-1 and adjusted p-value < 0.05 was applied. Gene ontology analysis were performed with DAVID software. For transcription factor analyses we implemented same analysis using the transcription factor target data from TRRUST2⁵⁰.”

Reviewer #3: Figure 4d-g,m should be in supplementary data (if included at all).

Our response: We have corrected these issues

Reviewer #3: Source of cell lines and of mice (especially transgenics) is not given. Why were littermates lacking the TRE-Myc not used as controls in place of wt FVB? Indeed, in Figure 2n, the text says WT mice, but the figure says ‘Myc Off’. Please check this.

Our response: We have now referenced {Shachaf, 2004 #254} for the mouse model source. These mice carry MYC as a repressible transgene integrated in the Y chromosome. Our breeding was set up so that all males were carrying the MYC transgene. The litters mates were females. The text was corrected to say WT mouse instead of MYC-OFF. We used WT males of the same age for better comparison, however we have also compared litter mate females and found that results were identical regardless of the gender of the control mice.

Reviewer #3: In methods, the immunohistochemistry section lacks several details that would be needed to allow reproduction of the experiments. Furthermore, no antibody information is given.

Our response: For IHC livers of WT or MYC-ON mice were fixed and stained as previously described using the anti MYC antibody ab32072 (Abcam) and anti Ki63 12202S (Cell Signaling) at 1:500 dilution¹³. Briefly, tissues were fixed in 10% formalin overnight, embedded in paraffin, and cut 5- μ m thick sections. The sections were stained with H &E, and immunohistochemistry performed for MYC and Ki-67. Tissues were

deparaffinized with xylene and ethanol. Antigen retrieval was done using the pressure cooker method and blocked with goat serum. Sections were incubated with the primary. Vectastain kit was used, and secondary antibody was used at a concentration of 1:500, developed with DAB and mounted with Permount mounting media. Sections were analyzed using the Hamamatsu Nanozoomer and NDP view software. .”

Reviewer #3: Please correct ‘acetylcarnitines’ to acylcarnitines in the text.

Our response: This section has been removed from the revised manuscript

Reviewer #3: nM/mg should presumably be nmol/mg on several axes. Furthermore AU appears on some axes – please clarify what this means.

Our response: Units have been updated to ng of metabolite per mg of protein to better align with the units used in the paper. We used AU for ¹³C-tryptophan metabolites because we did not have a standard for ¹³-C metabolites to set the curves.

Reviewer #3: When Trp was re-introduced to the diet of these mice, the tumors rapidly grew (Figure S2G-H) - this is not shown (only viability) – please rephrase. Also, include H as the panel title.

Our response: These changes have been made

Reviewer #3: Please improve the resolution of Figure S3A and increase font for p-value.

Our response: This correction was made.

REVIEWER COMMENTS

Reviewer #1 (Remarks to the Author):

All of my previous concerns have been addressed, and the manuscript is much-improved and ready for publication.

Reviewer #2 (Remarks to the Author):

In this revised manuscript, Venkateswaran et al. present new data showing that an underexplored aspect of tryptophan metabolism is responsible for HCC tumor growth in a MYC driven model. These new data are striking and build off recently published data that identified a new link between tryptophan metabolism and activation of the aryl hydrocarbon receptor (AHR) to suppress the immune system and drive tumor growth (Sadik et al Cell 2020). The authors demonstrated that the entire effect of the tryptophan free diet on tumor growth in vivo could be rescued by I3P supplementation. They also showed strong evidence of this in vitro suggesting a (large) portion of the effect was tumor cell dependent. The original manuscript had a compelling phenotype but this new mechanistic data significantly enhances the impact of the study. As the proposed mechanistic rationale has now shifted and the new hypothesis well supported by data, my major original concerns are either no longer valid or well-addressed.

Specifically, the authors extensively profiled protein synthesis and found no changes upon tryptophan restriction. This negative finding is supportive of their novel I3P mechanism and helps differentiate the observed effects from other studies that have investigated AA restriction and tumor growth. The authors have also removed data suggesting HFD could rescue tumor growth as it appears to act by a different mechanism than the I3P mechanism presented here.

As such I have no significant concerns and recommend this study for publication. There are figure panels with unclear or missing legends (Fig 8b is an example). I see no further experiments as necessary at this time, although I would be interested to see the results of a tumor growth experiment using HCC53N cells and siRNA or CRISPR against IL4I1.

Reviewer #3 (Remarks to the Author):

This revised manuscript from Venkateswaran et al. concerning Trp-dependence of MYC-driven liver tumorigenesis includes numerous data addressing concerns of the reviewers. The focus of the latter part of the original manuscript upon fatty acid oxidation has been removed and has been replaced by a more mechanistic insight into how Trp can support growth and proliferation in MYC-driven liver cancer cell lines. The authors also provide evidence that the Trp-derived metabolite I3P supports MYC-driven liver tumours in vivo. Overall, the manuscript now has a more concise message and, in terms of the Trp-derived metabolite I3P, is a timely addition to the field. The authors also show that Trp withdrawal leads to minimal changes in translation (several assays applied), which is an important finding in itself. It also appears as though Trp withdrawal may act, at least in part, through suppression of MYC activity (either levels or downstream transcriptional activity) – underlying mechanism(s) are unknown but, again, an important finding.

I have some concerns, detailed below, about the new data but I think that the manuscript is important and reveals targetable vulnerabilities for MYC-driven tumours that could translate into clinical application.

Major concerns

1. The phenotypic data provided about I3P rescue of growth/tumorigenesis in the absence of Trp are not accompanied by supporting metabolic/metabolomics data. Inclusion of such data is essential for a manuscript in which a single metabolite is of such central importance. In particular, although the authors provide evidence that I3P functions through the AhR receptor in their in vitro systems, corresponding data are lacking (indeed negative) in their in vivo studies, potentially invoking a model based upon other related metabolites.

In the in vivo experiments, I3P dietary rescue (without Trp) may provide I3P for gut bacteria to produce Trp (and other derivatives) and thereby support liver tumorigenesis (essentially a straight Trp replacement). Please report the effect of I3P administration in mice (with/without Trp in the diet) on the levels of Trp and Trp-related metabolites in liver (as in Fig 8b) and in serum. This is particularly important, given that I3P did not induce canonical AhR targets in liver.

As with the in vivo setting, I would request that the authors provide metabolic data indicating how I3P supplementation affects levels of Trp and all of its downstream relatives in the cell lines. It is entirely possible that alleviation of Trp usage through the I3P pathway (through I3P supplementation) may augment other Trp pathways (even in the absence of Trp in the medium – FCS contains Trp in polypeptides).

2. The transcriptional and growth-related effects of Trp starvation are, at least in part, due to reduction in MYC levels, as the authors mention. It is, therefore, important to know if the rescue effects of I3P administration (in vitro and in vivo) are due to restoration of MYC levels and, ideally, if AhR inhibition influences MYC levels or transcriptional activity (e.g. by measuring CAD/ODC or other canonical target mRNA levels or through RNAseq as in Fig 4I).

Minor concerns

Fig 1f and Fig 8b seem to show the opposite in liver tumours. 8b shows an elevation of kynurenine upon Myc activation, whereas 1f shows a strong reduction. From the Figure legend, it appears that Fig 8b may refer to Trp-starvation – If so, please make this clear in the text where the figure is cited and, ideally, on the figure panel itself.

Gene names should be italicised (e.g. MYC in 2nd para of introduction).

NAD⁺ and NADP⁺ should have superscript '+' throughout the manuscript.

Fig S4i needs a y-axis label.

Reviewer #3 (Remarks to the Author):

Reviewer #3: This revised manuscript from Venkateswaran et al. concerning Trp-dependence of MYC-driven liver tumorigenesis includes numerous data addressing concerns of the reviewers. The focus of the latter part of the original manuscript upon fatty acid oxidation has been removed and has been replaced by a more mechanistic insight into how Trp can support growth and proliferation in MYC-driven liver cancer cell lines. The authors also provide evidence that the Trp-derived metabolite I3P supports MYC-driven liver tumours in vivo. Overall, the manuscript now has a more concise message and, in terms of the Trp-derived metabolite I3P, is a timely addition to the field. The authors also show that Trp withdrawal leads to minimal changes in translation (several assays applied), which is an important finding in itself. It also appears as though Trp withdrawal may act, at least in part, through suppression of MYC activity (either levels or downstream transcriptional activity) – underlying mechanism(s) are unknown but, again, an important finding. I have some concerns, detailed below, about the new data but I think that the manuscript is important and reveals targetable vulnerabilities for MYC-driven tumours that could translate into clinical application.

Our response: We thank the reviewer for the feedback - we have diligently worked to address all comments through a combination of experimental modifications and detailed responses outlined below.

Major concerns

Reviewer #3: The phenotypic data provided about I3P rescue of growth/tumorigenesis in the absence of Trp are not accompanied by supporting metabolic/metabolomics data. Inclusion of such data is essential for a manuscript in which a single metabolite is of such central importance.

Our response: In response to the reviewer's feedback, we measure I3P levels and demonstrated that I3P is taken up by cancer cells in vitro and in vivo (**Figure 7k, Figure 7q and Figure 8j**).

Reviewer #3: In particular, although the authors provide evidence that I3P functions through the AhR receptor in their in vitro systems, corresponding data are lacking (indeed negative) in their in vivo studies, potentially invoking a model based upon other related metabolites.

Our response: To directly explore the impact of I3P on AHR signaling, we cultured liver cancer cells in the presence of I3P for 4, 8, and 16 hours. Our previous published studies demonstrated that it takes a minimum of 4 hours after ligand binding to detect quantifiable changes in the expression of AHR target genes. Our findings reveal that I3P promotes the expression of AHR canonical targets, including CYP1A1, NQO1, AHRR, and to a lesser extent UMP and SCIN. This provides evidence for the activation of AHR by I3P. This experiment is shown in **Figure 8m**. It is possible that AHR-independent functions of I3P exist and we will investigate these in the future by performing unbiased studies including RNA-seq and metabolomics in liver cancer cells that are WT or KO for AHR. Due to the poor quality of western blots obtained with antibodies for AHR target in liver tissues, we decided not to use those experiments for the paper. Specifically, the presence of several bands made it difficult to gain confidence were the correct bands. On the other hand, Western using cultured cell lysates provided very clean results.

Reviewer #3: In the in vivo experiments, I3P dietary rescue (without Trp) may provide I3P for gut bacteria to produce Trp (and other derivatives) and thereby support liver tumorigenesis (essentially a straight Trp replacement). Please report the effect of I3P administration in mice (with/without Trp in the diet) on the levels of Trp and Trp-related metabolites in liver (as in Fig 8b) and in serum. This is particularly important, given that I3P did not induce canonical AhR targets in liver. As with the in vivo setting, I would request that

the authors provide metabolic data indicating how I3P supplementation affects levels of Trp and all of its downstream relatives in the cell lines. It is entirely possible that alleviation of Trp usage through the I3P pathway (through I3P supplementation) may augment other Trp pathways (even in the absence of Trp in the medium – FCS contains Trp in polypeptides).

Our response: All I3P injections were administered intraperitoneally (IP) rather than through gavage, thereby minimizing exposure of the gut microbiome to the externally supplied I3P. Addressing concerns raised by the reviewer, we conducted experiments measuring Trp levels in mouse xenografts (**Figure 7q**) and MYC-on tumors (**Figure 8j**) extracted from animals subjected to a No-Trp diet supplemented with either vehicle or I3P injections. These investigations demonstrate that I3P injections did not elevate Trp levels *in vivo*. Additionally, we observed no alterations in Trp levels in cell lines cultured in either No-Trp media or complete media supplemented with I3P (**Figure 8k**). These results establish that, under the specific experimental conditions employed in this study, the administration of exogenous I3P does not impact Trp levels either *in vivo* or *in vitro*.

For cultured cells, we have quantified both Trp and I3P levels in HCC53N cells cultured with I3P. Currently, our laboratory is broadening these by conducting comprehensive metabolic and transcriptional studies to delineate the full spectrum of effects induced by I3P. This investigation is being expanded to encompass multiple liver and colon cancer cell lines, examining alterations in each of the Trp metabolism pathways. Moreover, we are using ¹³C Trp and tracing the flux of Trp into all metabolites under various conditions intra and extracellularly. Anticipating the complexity of these analyses, we foresee that these experiments will require a substantial amount of time to reach completion. We are committed to conducting thorough and rigorous assessments to ensure a comprehensive understanding of the impact of I3P on cellular processes in diverse cancer cell lines.

Reviewer #3: 2. The transcriptional and growth-related effects of Trp starvation are, at least in part, due to reduction in MYC levels, as the authors mention. It is, therefore, important to know if the rescue effects of I3P administration (*in vitro* and *in vivo*) are due to restoration of MYC levels and, ideally, if AhR inhibition influences MYC levels or transcriptional activity (e.g. by measuring CAD/ODC or other canonical target mRNA levels or through RNAseq as in Fig 4l).

Our response: This is an important question. We have clearly demonstrated that I3p incubation with liver cancer cells or I3P supplementation to MYC-ON mice starved of Trp have no effects on MYC levels. Please see Figures 8m, 8n for liver cancer cells and Figure S9 b for MYC-ON mice.

Minor concerns

Reviewer #3: Fig 1f and Fig 8b seem to show the opposite in liver tumours. 8b shows an elevation of kynurenine upon Myc activation, whereas 1f shows a strong reduction. From the Figure legend, it appears that Fig 8b may refer to Trp-starvation – If so, please make this clear in the text where the figure is cited and, ideally, on the figure panel itself.

Our response: We thank the reviewer for noticing the missing legend. And yes, Figure 8b compares metabolites in the liver of mice fed a control diet or No-Trp diet. This has been corrected.

Reviewer #3: Gene names should be italicised (e.g. MYC in 2nd para of introduction).

Our response: We have italicized “MYC” when referring to the gene. Thank you again for noticing this mistake.

Reviewer #3: NAD⁺ and NADP⁺ should have superscript ‘+’ throughout the manuscript.

Our response: These corrections have been made.

Reviewer #3: ig S4i needs a y-axis label.

Our response: The label was added to the y-axis.

REVIEWERS' COMMENTS

Reviewer #3 (Remarks to the Author):

I appreciate the authors' responses and have no remaining concerns. I look forward to seeing the manuscript published.